# Optimising supervised machine learning algorithms predicting cigarette cravings and lapses for a smoking cessation just-in-time adaptive intervention (JITAI)

Corinna Leppin[1]*, Jamie Brown[1], Claire Garnett[1,2], Dimitra Kale[1], Tosan Okpako[1], David Simons[3], Olga Perski[1,4]

1 Department of Behavioural Science and Health, University College London, London, United Kingdom, 2 School of Psychological Science, University of Bristol, Bristol, United Kingdom, 3 Department of Anthropology, The Pennsylvania State University, University Park, Pennsylvania, United States of America, 4 Department of Psychology, Stockholm University, Stockholm, Sweden

* corinna.leppin.21@ucl.ac.uk

## Abstract

This study aimed to optimise the balance between participant burden and algorithm performance for predicting high-risk moments in a smoking cessation just-in-time adaptive intervention (JITAI) by systematically varying ecological momentary assessment (EMA) prompt frequency, predictor count, and training data source. Thirty-seven participants completed 16 EMAs per day for the first 10 days of their smoking cessation attempt, reporting mood, context, behaviour, cravings, and smoking lapses. Random forest algorithms predicting lapses and cravings were evaluated in terms of F1-score and ROC-AUC via mixed effects models accounting for clustering within individuals. Performance across out-of-sample individuals ranged from excellent to poor but was, on average, modest. Lapse prediction outperformed craving prediction, particularly for ROC-AUC (Median F1-score: Lapses 0.436 [IQR 0.180–0.625], Cravings 0.400 [IQR 0.048–0.649]; Median ROC-AUC: Lapses 0.659 [IQR 0.514–0.809], Cravings 0.628 [IQR: 0.510–0.729]). A substantial proportion of configurations fell below commonly used minimum performance thresholds, particularly for F1-score. Reducing EMA frequency had outcome- and metric-dependent effects. Lapse F1-scores improved with fewer prompts (16 EMAs: 0.254 [IQR 0.081–0.500], 3 EMAs: 0.588 [IQR 0.353–0.667]), while ROC-AUC showed a slight, inconsistent decline (16 EMAs: 0.661 [IQR 0.520–876], 4 EMAs: 0.613 [IQR 0.494–0.786], 3 EMAs: 0.704 [IQR 0.567–0.809]). For cravings, both metrics declined with fewer prompts (F1-score: 16 EMAs: 0.470 [IQR 0.141–0.745]; 3 EMAs: 0.333 [IQR 0.000–0.600]; ROC-AUC: 16 EMAs 0.700 [IQR 0.582–0.811], 3 EMAs 0.544 [IQR 0.421–0.676]). Feature reduction had negligible impact on lapse prediction (F1-score: all features 0.435, selected features 0.441; ROC-AUC: all 0.660, selected 0.657), but slightly reduced craving performance (F1-score: all 0.410 [IQR

**Data availability statement:** The complete analysis syntax and data can be found on Github (https://github.com/OlgaPerski/EMASENS_ML/tree/main) and the Open Science Framework (https://osf.io/tnu72/).

**Funding:** This analysis is part of a project funded by Cancer Research UK (PRCRPG-Nov21\100002; https://www.cancerresearchuk.org/funding-for-researchers/our-funding-schemes/prevention-and-population-research-programme-awards). CG is supported by a National Institute for Health and Care Research (NIHR) Advanced Fellowship (302923; https://www.nihr.ac.uk/career-development/research-career-funding-programmes). OP is supported by HORIZON TMA MSCA Postdoctoral Fellowships - Global Fellowships from the European Union (Grant agreement number: 101065293; https://ec.europa.eu/info/funding-tenders/opportunities/portal/screen/opportunities/projects-details/43108390/101065293) and by the European Research Council (ERC-2025-StG) under Horizon Europe (HORIZON) (Grant Agreement number: 101219875; https://ec.europa.eu/info/funding-tenders/opportunities/portal/screen/opportunities/projects-details/43108390/101219875) Views and opinions expressed are however those of the author(s) only and do not necessarily reflect those of the European Union. The European Union cannot be held responsible for them. The funders had no role in study design, data collection and analysis, decision to publish, or preparation of the manuscript. There was no additional external funding received for this study.

**Competing interests:** OP and JB act as unpaid scientific advisors to the Smoke Free app. They do not receive salaries or any other remuneration for this work. CG, DK, CL, TO, and DS have no competing interests to declare.

0.117–0.646], selected 0.400 [IQR 0.000–0.650]; ROC-AUC: all 0.632, selected 0.622). Including participant-specific data improved lapse F1-scores (None 0.286 [IQR 0.000–0.571], 30 pc 0.542 [IQR: 0.329–0.667]), but not ROC-AUC (None 0.655 [IQR: 0.512–0.786], 30 pc 0.694 [IQR 0.513–0.852]); and impaired craving ROC-AUC (None 0.650 [IQR: 0.544–0.734], 30 pc 0.614 [IQR 0.493–0.730]; F1-score: None 0.424 [IQR 0.143–0.649], 30 pc 0.400 [IQR 0.000–0.703]). Overall, EMA-based machine learning detected lapse risk but showed modest overall performance and substantial inter-individual variability. Algorithms using higher EMA density, larger predictor sets, and participant-specific training data did not consistently outperform over more parsimonious approaches. However, machine learning prediction alone is unlikely to be sufficient for real-world JITAI implementation and may be best combined with complementary rules-based approaches.

## Introduction

Smoking remains a key cause of morbidity and mortality in the UK [1,2]. Improved interventions are needed to increase smoking cessation rates and improve population health in line with government targets [3]. Traditional behavioural interventions often focus on general motivation and coping strategies, but may be insufficient to prevent relapse [4,5]. Many people who smoke fail to actively seek out support at critical moments [6,7], and ensuing lapses often lead to full relapses [8–11]. Just-in-time adaptive interventions (JITAIs) aim to address this gap by delivering timely, tailored support during high-risk moments [12,13].

JITAIs rely on frequent assessment of tailoring variables (i.e., factors that may precipitate or shape high-risk moments), actively via ecological momentary assessments (EMAs) or passively via sensors. Decision rules (i.e., algorithms that specify how to combine data from tailoring variables) are then used to determine opportune moments for intervention delivery [13]. While early evidence suggests smoking cessation JITAIs are feasible [14–19] and can be effective [18], there remain questions over how to optimally balance the two (potentially conflicting) requirements of JITAIs: collecting enough data on psychological and contextual tailoring variables to effectively detect or predict high-risk moments and receptivity and creating a monitoring regime that is not too burdensome for users. Balancing these two requirements can be challenging in practice, as, all else being equal, an algorithm's ability to detect or predict states of need and receptivity, decipher their nature, and provide appropriate support is usually (but not always, cf. [20]) improved by more data and more sophisticated computing techniques. However, the time and effort involved in data collection and the drain on device battery resulting from passive sensors and complex modelling can be a burden for users [21,22].

The minimum sampling frequency at which psychological and contextual tailoring variables need to be assessed to collect data able to predict or detect high-risk moments well enough to prevent smoking lapses and effectively support smoking cessation is unclear. Given the frequency of smoking behaviour and the rapidity at which predictor variables may provoke lapses, hourly or more frequent

measurements may be necessary to accurately model smoking dynamics [23,24]. However, previous research indicates that frequent or lengthy EMAs may reduce compliance and lead to disengagement with the intervention [25,26]. Although intensive data collection can enhance algorithm performance, most users are unlikely to tolerate more than four to five surveys with around 10 items per day for a prolonged period of time [16,19,21,27–30]. Limiting EMA frequency and item count per EMA may therefore be key to making a JITAI acceptable and maintaining user engagement.

JITAIs that are adaptive from the first interaction by using group-level algorithms, or gradually personalised "warm start" hybrid-level algorithms, appear to align better with user preferences than JITAIs that require extensive pre-quit training before providing tailored support [31,32]. While some previous evidence suggests that hybrid-level algorithms may be better at predicting high-risk moments during smoking cessation [9,29,33], they also require iterative retraining of the algorithms, which increases computational demand (and, particularly if done device-side, drain device battery, and if done server-side, have privacy risks) and adds complexity to the implementation process. This can limit their feasibility and acceptability, so a careful trade-off with performance compared to group-level algorithms is needed. Additionally, this is a relatively new area of research, and there is little guidance on how long a warm-start period should be, and how the target behaviour, the dynamics of the outcome, and the intended intervention context influence the optimal length of this warm start period.

The current study aimed to train and test a random forest algorithm for a future smoking cessation JITAI. A previously published proof-of-concept study used a series of supervised machine learning algorithms to demonstrate that smoking lapses and cravings can be accurately predicted using high-density EMA and sensor data [34]. This study extends that work using the same data, focusing on real-world implementation by systematically examining trade-offs between data collection burden and model performance. It explores how reducing the sampling frequency and the number of variables and varying the proportion of the test participants' own data in the training set affects the algorithm's ability to predict high-risk moments, operationalised either as lapses or as instances of high cravings and predicted in separate algorithms, during a smoking cessation attempt. The aim is to evaluate how prediction models can be adapted when data density is reduced to levels that are feasible and sustainable in practical clinical or public health settings.

The specific research question is: How does the performance of random forest algorithms predicting high-risk moments among people during the first 10 days of their smoking cessation attempt differ according to the frequency of EMA prompts per day, the number of predictor variables, the proportion of the test participant's own data in the training set, and whether high-risk moments are operationalised as smoking lapses or cigarette cravings?

## Materials and methods

### Design

This study comprised secondary analyses of previously collected EMA data from 37 people who smoke during the first 10 days of their quit attempt [34]. The sample size was based on a simulation informed by the available literature; for more details, see https://osf.io/ywqpv. Participants received 16 EMAs per day (one every waking hour) to assess their mood, pain level, activity, social and physical context, cigarette craving, caffeine, alcohol, and nicotine consumption, and lapse incidence.

Ethical approval was obtained from the UCL Research Ethics Committee (Project ID: 15297.004). Participants were asked for their written informed consent to share their anonymised research data with other researchers via an open science platform. The protocol for this study was preregistered on the Open Science Framework prior to the start of analyses (https://osf.io/tnu72/).

### Recruitment and participants

Participants were recruited between May 2022 and March 2023, mostly through paid adverts on Facebook and Instagram. Recruitment commenced on 20 May 2022, with the first participant's baseline visit taking place on 27 May 2022. The last participant had their baseline visit on 21 March 2023 and their final follow-up visit on 31 March 2023.

Eligibility requirements included being at least 18 years of age, regularly smoking cigarettes, residing in or around London, owning a smartphone, and being willing to set a quit date within seven days of the initial study visit. Smoking cigarettes regularly was defined as responding "Yes" to the question "Do you smoke cigarettes at all nowadays?" and at least one (1) to the question "How many cigarettes per day do you usually smoke?". Smoking status was not bio-verified at baseline. Participants were remunerated a minimum of £20 for participation, regardless of their EMA completion rate. Participants received £50 if they completed at least 50% of EMAs on at least 50% of study days and were informed they would earn an additional £5 for each day their EMA compliance exceeded 70%, up to a maximum of £100. It was made clear that payment was not tied to their smoking behaviour.

## Procedure

After completing an online screening survey to confirm eligibility, eligible participants were invited to attend an in-person baseline visit. At the baseline visit, they completed a baseline survey, provided written informed consent, and downloaded and received instructions for how to use the m-Path app [35], through which they received the EMAs throughout the study period. Participants were asked to set a quit date within the next seven days and encouraged to set it for the following day to capitalise on their motivation to stop smoking. Participants also received free access to the premium version of an evidence-based smoking cessation app [36]. At the end of the baseline visit, the in-person follow-up visit was scheduled.

The 10-day study period began on each participant's selected quit date. During this time, participants received 16 EMA prompts per day (one every waking hour on the hour or half-hour) via the m-Path app (signal-contingent sampling), tailored to their usual waking hours. EMAs needed to be completed within 30 minutes of receipt. Participants were also instructed to log lapses as they occurred using an in-app button (event-contingent sampling). Each EMA took around one to three minutes to complete and contained the same 22–25 items (depending on whether participants chose to add additional items; these additional items are not considered in this analysis, but more details on them can be found in the previous study using these data [34]). At the follow-up visit, data from study-related apps were exported, the m-Path app was deactivated, and participants received payment for their participation. Participants who reported that they had successfully quit smoking were asked to provide expired carbon monoxide measures using a Bedfont iCOquit smokerlyzer (https://www.icoquit.com/) to verify their abstinence (cut-off set at ≤8 parts per million). Full details on recruitment and procedure are reported in the previous study [34].

## Measures

Overall, the machine learning algorithms in this study used data from 34 variables: 12 between-person variables collected at the baseline questionnaire, and 22 within-person variables collected at every EMA. Full questionnaires and variable lists cand be found in S1 Appendix.

The between-person variables were collected at baseline and comprised two numerical variables (cigarettes per day; age in years), four binary variables (gender [four levels were available, but all participants selected either "man" or "woman"); post-16 education; ever-use of behavioural support for smoking reduction or cessation; ever-use of pharmacological support for smoking reduction or cessation), and six categorical variables (occupation type; ethnicity; past quit attempt; time to first cigarette; motivation to stop scale [37]; season of data collection). The within-person variables comprised 14 quasi-continuous variables measured on 11-point Likert scales (ten variables measuring affect informed by the circumplex model of affect [38–40]: sad, irritable, stressed, anxious, bored, calm, content, happy, excited, enthusiastic; bodily pain; cigarette cravings; momentary motivation not to smoke; quitting-related self-efficacy), four binary variables (past-hour caffeine consumption; past-hour alcohol consumption; past-hour nicotine use other than tobacco use/smoking; past-hour lapse [i.e., smoking), and four categorical variables (social context; activity; location; cigarette availability). For training and evaluating the algorithms, categorical variables were recoded into dummy variables.

In prior formative work using these data, we explored more complex representations of lapses and other time-varying predictors, including variables that incorporated longer-term smoking history and cumulative consumption or moving averages over extended windows. These more complex features did not meaningfully improve algorithm performance relative to the simpler representations used here and were therefore not pursued further in the present analyses.

## Analyses

**Data pre-processing.** We used the pre-processed dataset from the previous study [34]. In this pre-processed dataset, eight out of 46 participants with insufficient EMA adherence (defined as completion of ≤60% of prompts) were excluded. This adherence threshold was applied to ensure sufficient data density to support reliable estimation of temporal associations and the training and evaluation of machine-learning models relying on lagged EMA predictors. This criterion was also retained for models using fewer prompts/observations, as these were derived from the same intensive EMA protocol and still required adequate coverage across days and contexts. Including participants only in reduced-prompt conditions but not in full-prompt models would have created validity issues in the imputation process, introduced additional selection biases, and reduced comparability across algorithm specifications. Missing data in the remaining analytical sample were imputed using the univariate Kalman filter, which is recommended for univariate time series data imputation [41] and has been previously used in analysis of intensive longitudinal data on health behaviours [42]. As a forward filter that only uses past and current information and does not consider future states, it is also suitable for implementation within an eventual JITAI.

**Model specification.** This study used group- and hybrid-level random forest (RF) algorithms to predict (a) lapses by or (b) cravings (dichotomised; high ≥ 7 on an 11-point Likert scale from 0 to 10) at one EMA based on levels of the predictor variables at the previous EMA and the values of the baseline variables. The two outcomes were predicted using separate algorithms. RF algorithms are a machine learning method that combines the results of many decision trees to make more accurate and reliable predictions [43]. A decision tree is a flowchart-like model that splits the data into branches based on values of the predictor variables, aiming to group similar outcomes together [43]. In an RF, each tree is built using a random sample of the data and a random subset of predictors, which helps reduce overfitting and improves the model's ability to generalise to new data [43]. For every outcome by number of EMAs by set of predictors combination, the number of candidate predictors at each split and the minimum number of observations in a terminal node was tuned at the group-level, while the number of trees in the forest was set at 500, in line with recommendations for training RF algorithms [44–46]. As is standard, the classification threshold was set at 0.5, such that predicted probabilities ≥0.5 were classified as positive cases/events (lapse or high cravings) and probabilities < 0.5 as negative cases/non-events (non-lapses or low/no cravings). All analyses were conducted using R [47], and the models were trained and tested using the tidymodels framework [48]. The complete analysis syntax and data can be found on Github (https://github.com/OlgaPerski/EMASENS_ML/tree/main) and the Open Science Framework (https://osf.io/tnu72/).

**Selection of algorithm types.** As the aim of this study was to train and test an algorithm for use in a future smoking cessation JITAI, we tested a single algorithm type for pragmatic reasons. The choice of algorithm architecture was informed by a previous study using the same dataset [34], in which multiple algorithm architectures (RF, Support Vector Machine, Penalised Logistic Regression, and XGBoost) were systematically compared. In that earlier work, an RF architecture showed the most favourable balance performance across area under the receiver operating characteristic curve (ROC-AUC; overall ability to distinguish between classes), sensitivity (true positive rate), and specificity (true negative rate). Additionally, RF algorithms have modest computational demands, are well suited for feature selection, do not make assumptions about the shape of the relationship between predictors and outcomes, and can handle a mix of continuous and categorical predictors as well as complex interactions [43,49–52]. Therefore, we decided to take RF algorithms in this study and did not re-compare different algorithm classes. Instead, we held the algorithm type constant (RF) and examined how different *specification decisions* (described in more detail in the following paragraphs) affected performance.

**Number and timing of prompts.** We fit algorithms based on the data from 3, 4, 5, 6, and all 16 prompts per day. With a view to JITAI implementation considerations, we used quasi-random prompts evenly spaced throughout the day. To do this, we randomly selected prompts from blocks within the participant's daily schedule. The number of prompts to be tested was based on previous literature [16,19,21,27–30,53]. The prompts chosen for each number of prompts per day were as follows:

- 3: randomly choose one of 1–5, 6–11, 12–16

- 4: randomly choose one of 1–4, 5–8, 9–12, 13–16

- 5: randomly choose one of 1–3, 4–6, 7–9, 10–13, 14–16

- 6: randomly choose one of 1–3, 4–6, 7–8, 9–11, 12–14, 15–16

- All 16 prompts

All event-based EMA reports (i.e., when a participant reported smoking outside of a prompt) were included in all algorithms, regardless of the number of prompts per day tested.

**Outcome definitions.** The different outcomes, lapses and high cravings, were predicted using separate algorithms. We defined lapses as a lapse event that occurred between a participant's current and the next randomly selected/event-based EMA. The EMA was counted as a lapse if any lapse event was observed between these prompts. For high cravings, we recorded whether a high craving event was reported at the next randomly selected prompt. Additionally, we calculated prior lapses (as a predictor) by checking if a lapse occurred between the previous and current selected prompts.

High cravings were derived from an 11-point Likert cale (0–10) measuring self-reported craving, using a cut-off of ≥7. Cravings were dichotomised to align the modelling approach with the intended downstream use of the algorithm as part of a JITAI, where the primary aim is to identify discrete high-risk moments at which to deliver support. The cut-off of ≥7 was chosen for two complementary reasons. Firstly, it is consistent with the cut-off used in the previous analysis of the same dataset [34], facilitating comparability across studies and allowing interpretation of results within an established analytic framework. Secondly, a threshold of ≥7 captured approximately one third of craving observations across prompt frequencies (more details below), providing a pragmatic balance between identifying clinically meaningful high-craving states, maintaining sufficient class prevalence for stable model training and evaluation, and balancing burden with responsiveness.

In line with the previous study [34] and to allow for the calculation of the performance metrics, only participants with at least one observation in each outcome class (i.e., lapse and non-lapse for lapse-prediction algorithms, and high craving and low/no craving for craving-prediction algorithms) were included. When constructing hybrid-level algorithms, participants were additionally required to have at least one observation in each outcome class in the testing set.

**Number of predictors.** We used data-driven feature selection to reduce the number of predictors. Feature selection describes machine learning techniques that aim to reduce the number of input dimensions while preserving the algorithm's performance [54]. In smoking cessation, filter [55] and embedded [30] feature selection methods have been previously used to improve the parsimony of models predicting smoking urges and lapses. The feature selection technique we used is recursive feature elimination with cross-validation (RFE-CV). RFE is an embedded feature selection method that is well-suited in this case, as it performs well even in the presence of correlated predictors [56], can handle both continuous and categorical predictors together and structured features, and does not have the same bias towards high cardinality features as some other methods. RFE-CV selects an optimal number of features without the need for pre-specification [57] and has been shown to have fairly high stability [58]. RFE-CV has shown promise in the epidemiology and healthcare fields [59–61]. To compare algorithms with fewer predictors to those with the full set of 57 predictors, we compared

the performance of algorithms using the optimal set of predictors to that of algorithms with the full set of predictors. In cases where the optimal set of predictors was the full set, we compared the performance of algorithms using the top 15 predictors as identified by RFE-CV with that of algorithms using the full set of predictors. The full list of predictors used for each algorithm is provided in S2 Appendix.

**Proportion of own data in the training set.** For this study, group-level algorithms and hybrid-level algorithms were used to predict lapses and high cravings, with the latter incorporating the initial day or days of the test participant's own data, along with the data from the other participants, in the training set. Including a participant's data in the training set means using a portion of their data to help the machine learning model learn patterns during the training phase, before generating predictions on the participant's later (unseen) data, which is reserved for testing the model's performance. With a view to how a JITAI would be implemented in a real-world setting and to retain adequate amounts of data in the testing set, the specific amount of the participant's own data in the training set was varied as 10% (first day), 20% (first two days), and 30% (first three days).

**Performance.** Algorithm performance was evaluated using the F1-score and ROC-AUC as primary metrics with sensitivity and specificity reported as supplementary metrics. All metrics were calculated using a fixed probability threshold of 0.5 to convert predicted probabilities into binary class predictions.

The F1-score summarises binary classification performance as the harmonic mean of precision and recall (sensitivity), balancing false positives and false negatives: F1-score = 2 × (precision × recall)/ (precision + recall)

Precision (positive predictive value) is defined as: Precision = True Positives/ (True Positives + False Positives)

Recall (sensitivity) is defined as: Recall = True Positives/ (True Positives + False Negatives)

F1-scores range from 0 to 1, with higher values indicating better performance. Scores below 0.5 are generally not seen as acceptable. Because the F1-score penalises large imbalances between precision and recall, the F1-score is well suited to imbalanced data.

ROC-AUC reflects the model's overall ability to discriminate between positive and negative classes across all possible thresholds. It can be interpreted as the probability that a randomly selected positive observation receives a higher predicted risk than a randomly selected negative observation. Values range from 0 to 1 with 0.5 indicating chance-level (and therefore unacceptable) discriminatory ability and 1 indicating perfect discrimination.

Sensitivity and specificity quantify performance at the chosen threshold of 0.5.

Sensitivity (or recall, as mentioned above) is defined as: True Positives/ (True Positives + False Negatives)

Specificity is defined as: True Negatives/ (True Negatives + False Positives)

Sensitivity reflects the ability to detect high-risk moments, whereas specificity reflects correct identification of low-risk moments. For cases such as this, where the detection of high-risk moments takes precedence, a minimum sensitivity of 0.7 and a minimum specificity of 0.5 are generally considered the boundaries for acceptable performance.

To examine patterns in performance metrics, we used generalised linear mixed models with link functions appropriate for the distribution of each metric, implemented using the glmmTMB and lme4 packages in R [62–64]. Based on distributional analysis, we used linear mixed models for ROC-AUC (approximately normally distributed), zero-inflated beta regression for F1-score (bounded continuous outcome with substantial proportions of zeros), and zero-one-inflated beta regression for sensitivity and specificity (bounded continuous outcomes with substantial proportions of boundary values at both zero and one). Models included random intercepts for each participant to account for within-subject clustering, and fixed effects for the number of prompts per day (EMA frequency), outcome type (lapses, cravings), use of feature selection (yes, no), and the proportion of own data in the training set (none, 10%, 20%, 30%). Separate models were fitted for each performance metric across outcomes, and stratified by outcome (cravings or lapses), with outcome excluded as a fixed effect in the stratified models. For fixed effects with more than two levels, marginal effects were estimated using the emmeans package [65] to compute pairwise comparisons with Tukey's adjustment for multiple comparisons.

## Changes from pre-registration

Two changes were made to the pre-registered protocol. The first change was the addition of group-level algorithms. Originally, we only planned to train and evaluate hybrid-level algorithms. However, after conducting the initial analyses, we realised that there were a substantial number of participants who only had lapses in the first day(s) of their quit attempt, and for whom, consequently, hybrid-level algorithms could not be constructed, and saw that the performance of hybrid-level algorithms appeared unrelated to the amount of the participant's own data in the training set. Therefore, we chose to additionally train and evaluate group-level algorithms and to include the proportion of own data in the training set as its own research question rather than as a sensitivity analysis. The second change was related to the method of feature selection. Originally, we planned to only compare the optimal set of predictors as determined by the RFE-CV algorithms with the full set of predictors. However, for a substantial number of algorithm configurations, the optimal set of predictors was all predictors. As that result would not allow us to compare the performance of algorithms with fewer predictors with that with the full set of predictors, we chose to select the 15 most important predictors as indicated by the RFE-CV procedure if that was the case and compare the algorithm with only these predictors with that with the full set of predictors.

Additionally, the preregistration did not specify a primary performance metric, but we focused on ROC-AUC and the F1-score in the main results for clarity and brevity. Additionally, the preregistration did not specify how patterns in performance metrics would be analysed. Given the large number of algorithm configurations (5 prompt frequencies × 2 outcomes × 2 feature selection levels × 4 levels of the test participant's own data in the training set, each applied to every out-of-sample individual), we used mixed models appropriate for the distribution of the respective score to summarise performance. This approach offers a more interpretable and statistically robust overview than could be achieved through visual inspection or comparisons of specific algorithm configurations alone.

# Results

## Sample description

The analytic sample comprises n = 37 adult people who smoke from in and around London during the first 10 days of their quit attempt. Initial pre-processing had left a sample of n = 38 participants, but one further participant was excluded from the analytic sample after because they had no instances of either high cravings or lapses and therefore did not meet the requirement of having at least one observation of each outcome class in each outcome set. Table 1 outlines participant characteristics.

The majority of participants were female (62.2%) and of any White ethnic background (81.1%). The mean age was 42.97 years (Standard deviation (SD) = 14.45). Prior to quitting, participants smoked an average of 13.08 cigarettes per day (SD = 6.50). At follow-up, 43.2% had smoked in the past 24 hours, 10.8% smoked 1–6 days ago, and 32.4% had not smoked since their planned quit date. The average adherence to the EMAs was 76.86% (SD = 9.27).

Among participants with at least one lapse (n = 25; 67.6% of the entire analytics sample), demographic and behavioural characteristics were similar: 60.0% were female and 76.0% identified with any White ethnicity. Their mean age was 42.48 years (SD = 13.80), and they smoked an average of 13.48 cigarettes per day (SD = 6.72). Despite reporting lapses during their EMAs, 16.0% reported having remained abstinent since their planned quit date at the follow-up visit. The average EMA adherence was 76.38% (SD = 9.23).

Algorithms using a proportion of the participants' own data in the training set had lower sample sizes as only participants with at least one observation in each outcome class in the testing set were included. For algorithms predicting lapses, algorithms using 10% of participants' own data used data from 20 participants, while algorithms using 20% and 30% of participants' own data each used data from 17 participants. For algorithms predicting cravings, algorithms using 10% of participants' own data used data from all 37 participants, while algorithms using 20% and 30% of participants' own data each used data from 36 participants.

**Table 1. Participant characteristics.**

| Characteristic | All participants<br>N = 37[1] | Participants with at least one lapse<br>N = 25[1] |
|---|---|---|
| Age | 42.97 (14.45) | 42.48 (13.80) |
| Gender | | |
| Female | 23 (62.2%) | 15 (60.0%) |
| Male | 14 (37.8%) | 10 (40.0%) |
| Occupation | | |
| Non-manual | 17 (45.9%) | 12 (48.0%) |
| Manual | 9 (24.3%) | 5 (20.0%) |
| Other (e.g., student, unemployed, retired) | 11 (29.7%) | 8 (32.0%) |
| Post-16 educational qualifications | 33 (89.2%) | 22 (88.0%) |
| Ethnicity | | |
| Asian or Asian British (any Asian background) | 3 (8.1%) | 3 (12.0%) |
| Black, Black British, Caribbean or African (any Black, Black British or Caribbean background) | 1 (2.7%) | 0 (0.0%) |
| Mixed or multiple ethnic groups (e.g., White and Black African, White and Asian) | 2 (5.4%) | 2 (8.0%) |
| Other ethnic group (i.e., Arab) | 1 (2.7%) | 1 (4.0%) |
| White (any White background) | 30 (81.1%) | 19 (76.0%) |
| Cigarettes per day | 13.08 (6.50) | 13.48 (6.72) |
| Time to first cigarette | | |
| Within 5 minutes | 9 (24.3%) | 7 (28.0%) |
| 6–30 minutes | 16 (43.2%) | 10 (40.0%) |
| 31–60 minutes | 6 (16.2%) | 4 (16.0%) |
| After 60 minutes | 6 (16.2%) | 4 (16.0%) |
| Motivation to stop | | |
| I don't want to stop smoking | 0 (0.0%) | 0 (0.0%) |
| I think I should stop smoking but don't really want to | 2 (5.4%) | 2 (8.0%) |
| I want to stop smoking but haven't thought about when | 6 (16.2%) | 5 (20.0%) |
| I really want to stop smoking but don't know when I will | 7 (18.9%) | 3 (12.0%) |
| I want to stop smoking and hope to soon | 9 (24.3%) | 5 (20.0%) |
| I really want to stop smoking and intend to in the next 3 months | 2 (5.4%) | 2 (8.0%) |
| I really want to stop smoking and intend to in the next month | 11 (29.7%) | 8 (32.0%) |
| Past-year quit attempt | | |
| No, never | 3 (8.1%) | 2 (8.0%) |
| Yes, but not in the past year | 16 (43.2%) | 9 (36.0%) |
| Yes, in the past year | 18 (48.6%) | 14 (56.0%) |
| Ever use of pharmacological support (e.g., nicotine replacement therapy, varenicline, e-cigarettes) | 30 (81.1%) | 20 (80.0%) |
| Ever use of behavioural support (e.g., counselling, telephone support, website, app) | 16 (43.2%) | 10 (40.0%) |
| Smoked since quit date | | |
| In the last 24hrs | 16 (43.2%) | 14 (56.0%) |
| 1–6 days ago | 4 (10.8%) | 3 (12.0%) |
| 1–3 weeks ago | 5 (13.5%) | 4 (16.0%) |

*(Continued)*

**Table 1.** (Continued)

|  | All participants | Participants with at least one lapse |
|---|---|---|
| Have not smoked since my planned quit date | 12 (32.4%) | 4 (16.0%) |
| % Completed EMAs | 76.86 (9.27) | 76.38 (9.23) |

[1]Mean (standard deviation); n (%); percentages may not always add up to 100% due to rounding.

## EMA description

Participants in the full analytic sample completed a total of 5,964 EMAs. Table 2 summarises key descriptives across the different prompt frequencies (16, 6, 5, 4, and 3 prompts per day). Versions of this table restricted to participants with at least one lapse and to participants included in the different algorithm configurations using a subset of the test participant's own data in the training set are available in S3 Appendix.

Event-contingent EMAs represented a small fraction of the total, ranging from 0.7% (16 prompts per day) to 4.0% (3 prompts per day). Lapse rates varied by prompt frequency, from 6.9% (16 prompts per day) to 20.8% (3 prompts per day). Due to the difference in calculation, the proportion of high cravings was higher and more consistent, ranging from 35.8% (16 prompts per day) to 39.9% (3 prompts per day). The proportion of EMAs recording lapses and high cravings varied substantially across individuals.

## Overall algorithm performance

Across all algorithm specifications, performance varied markedly between individuals and was centrally modest. For lapse prediction, the median F1-score was 0.436 (mean 0.409; IQR 0.180–0.625), and the median ROC-AUC was 0.659 (mean 0.652; IQR 0.514–0.809). For craving prediction, the median F1-score was 0.400 (mean 0.410; IQR 0.048–0.649), and the median ROC-AUC was 0.628 (mean 0.611; IQR 0.510–0.729). A substantial proportion of configurations fell below commonly used minimum thresholds for acceptable performance, particularly when evaluated using the F1-score (see S4 Appendix to S7 Appendix for details per metric and configuration). A full summary of key performance measures by predictors, algorithm specifications, and participants can be found in S4 to S7 Appendices.

## Research question 1: Prompt frequency

**Lapses.** Counter to the expectation that denser sampling improves performance, median F1-scores increased monotonically as prompt frequency decreased for lapse prediction: 16 prompts/day (median 0.254; mean 0.308; IQR

**Table 2.** Description of the observations.

| Prompts per day | EMAs | | | | Lapses | | | | High cravings | | | |
|---|---|---|---|---|---|---|---|---|---|---|---|---|
| | Number | | | Propor-tion event contingent | Overall proportion | Across individuals | | | Overall proportion | Across individuals | | |
| | Total | Event contingent | | | Signal contingent | | Median proportion | 25th percentile proportion | 75th percentile proportion | Median proportion | 25th percentile proportion | 75th percentile proportion |
| 16 | 5,964 | 44 | 5,920 | 0.7% | 6.9% | 1.2% | 0.0% | 7.5% | 35.8% | 29.6% | 11.1% | 52.8% |
| 6 | 2,204 | 44 | 2,160 | 2.0% | 12.9% | 3.3% | 0.0% | 18.2% | 37.3% | 30.9% | 15.3% | 59.4% |
| 5 | 1,742 | 42 | 1,700 | 2.4% | 15.2% | 3.0% | 0.0% | 26.0% | 38.1% | 29.5% | 16.9% | 53.7% |
| 4 | 1,401 | 41 | 1,360 | 2.9% | 17.0% | 3.7% | 0.0% | 28.8% | 40.1% | 29.8% | 17.5% | 57.7% |
| 3 | 1,063 | 43 | 1,020 | 4.0% | 20.8% | 6.6% | 0.0% | 32.3% | 39.0% | 31.2% | 16.3% | 55.6% |

0.081–0.500), 6 prompts (median 0.440; mean 0.400; IQR 0.222–0.587), 5 prompts (median 0.444; mean 0.397; IQR 0.143–0.573), 4 prompts (median 0.475; mean 0.432; IQR 0.218–0.640), and 3 prompts/day (median 0.588; mean 0.508; IQR 0.353–0.667) (Table 3).

Zero-inflated beta-regressions of the scores confirmed this pattern: relative to 16 prompts/day, F1-scores were significantly higher at lower prompt frequencies (all p < 0.001) (Table 4). All pairwise contrasts between 16 prompts/day and lower frequencies were statistically significant after adjustment (S8 Appendix).

In contrast, ROC-AUC showed slightly poorer performance with reduced prompting. Median ROC-AUCs were similar across conditions (16: median 0.661, mean 0.680, IQR 0.520–0.876; 6: median 0.628, mean 0.637, IQR 0.498–0.791;

**Table 3. Summary of F1-score by algorithm configurations (lapses).**

| | Number of algorithms | Median | Mean | Standard deviation | 25th percentile | 75th percentile | Minimum | Maximum |
|---|---|---|---|---|---|---|---|---|
| Lapses overall | | | | | | | | |
| | 780 | 0.436 | 0.409 | 0.283 | 0.180 | 0.625 | 0.000 | 0.986 |
| Prompts per day | | | | | | | | |
| 16 | 158 | 0.254 | 0.308 | 0.274 | 0.081 | 0.500 | 0.000 | 0.857 |
| 6 | 156 | 0.440 | 0.400 | 0.262 | 0.222 | 0.587 | 0.000 | 0.945 |
| 5 | 156 | 0.444 | 0.397 | 0.277 | 0.143 | 0.573 | 0.000 | 0.955 |
| 4 | 156 | 0.475 | 0.432 | 0.285 | 0.218 | 0.640 | 0.000 | 0.986 |
| 3 | 154 | 0.588 | 0.508 | 0.281 | 0.353 | 0.667 | 0.000 | 0.962 |
| Feature selection | | | | | | | | |
| All Features | 390 | 0.435 | 0.404 | 0.278 | 0.188 | 0.620 | 0.000 | 0.986 |
| Selected Features | 390 | 0.441 | 0.413 | 0.287 | 0.168 | 0.632 | 0.000 | 0.986 |
| Proportion of own data in training set | | | | | | | | |
| None | 242 | 0.286 | 0.319 | 0.286 | 0.000 | 0.571 | 0.000 | 0.963 |
| 10% | 198 | 0.400 | 0.384 | 0.282 | 0.155 | 0.607 | 0.000 | 0.986 |
| 20% | 170 | 0.500 | 0.484 | 0.238 | 0.308 | 0.650 | 0.000 | 0.984 |
| 30% | 170 | 0.542 | 0.490 | 0.278 | 0.329 | 0.667 | 0.000 | 0.982 |

**Table 4. Zero-inflated beta-regression model (with random intercepts) of F1-Score (lapses).**

| Predictor | Estimate (logit scale) | Standard error | 95% confidence interval (logit) | z-statistic | p-value | Sig. |
|---|---|---|---|---|---|---|
| Intercept | −0.785 | 0.203 | −1.183, −0.386 | −3.859 | < 0.001 | *** |
| Prompts per day: 6 | 0.460 | 0.075 | 0.313, 0.606 | 6.162 | < 0.001 | *** |
| Prompts per day: 5 | 0.411 | 0.074 | 0.266, 0.557 | 5.537 | < 0.001 | *** |
| Prompts per day: 4 | 0.655 | 0.075 | 0.508, 0.801 | 8.753 | < 0.001 | *** |
| Prompts per day: 3 | 0.981 | 0.076 | 0.833, 1.130 | 12.957 | < 0.001 | *** |
| Feature selection: Selected features | 0.097 | 0.047 | 0.006, 0.188 | 2.079 | 0.038 | * |
| Share of own data: 10% | 0.091 | 0.065 | −0.037, 0.219 | 1.395 | 0.163 | |
| Share of own data: 20% | 0.198 | 0.065 | 0.072, 0.325 | 3.071 | 0.002 | ** |
| Share of own data: 30% | 0.384 | 0.067 | 0.253, 0.514 | 5.767 | < 0.001 | *** |

Reference levels: Prompts per day = 16; Feature selection = All features; Share of own data = None.

Note: Estimate coefficients are on the logit scale (conditional component only).

This is a zero-inflated beta model with separate processes for zeros and values between 0 and 1.

For interpretable effect sizes (actual differences in predicted values), see pairwise comparisons on the response scale (0–1) in S8 Appendix.

Significance levels: *** p < 0.001, ** p < 0.01, * p < 0.05.

5: median 0.660, mean 0.646, IQR 0.510–0.795; 4: median 0.613, mean 0.631, IQR 0.494–0.786; 3: median 0.704, mean 0.665, IQR 0.567–0.809) (Table 5). Linear mixed models indicated significantly lower ROC-AUC at 6, 5, and 4 prompts/day compared with 16 (p-values between.003 and <.001), while 3 prompts/day did not differ significantly from 16 (p = .226) (Table 6). See S9 Appendix for full pairwise comparisons.

**Cravings.** For craving prediction, reduced prompt frequency was associated with consistent performance declines. Median F1-scores decreased from 0.470 (mean 0.465; IQR 0.141–0.745) at 16 prompts/day to 0.447 (mean 0.453; IQR 0.211–0.658) at 6 prompts, 0.368 (mean 0.393; IQR 0.000–0.624) at 5 prompts, 0.361 (mean 0.379; IQR 0.000–0.647) at 4 prompts, and 0.333 (mean 0.358; IQR 0.000–0.600) at 3 prompts/day (Table 7). Zero-inflated beta-regressions indicated significantly lower F1-scores for all reduced-prompt conditions relative to 16 prompts/day (all p ≤ .012) (Table 8).

ROC-AUC showed a similar pattern. Median ROC-AUC declined from 0.700 (mean 0.681; IQR 0.582–0.811) at 16 prompts/day to 0.652 (mean 0.641; IQR 0.575–0.736), 0.606 (mean 0.598; IQR 0.517–0.696), 0.622 (mean 0.593; IQR

**Table 5. Summary of ROC-AUC by algorithm configurations (lapses).**

| | Number of algorithms | Median | Mean | Standard deviation | 25th percentile | 75th percentile | Minimum | Maximum |
|---|---|---|---|---|---|---|---|---|
| Lapses overall | | | | | | | | |
| | 780 | 0.659 | 0.652 | 0.198 | 0.514 | 0.809 | 0.000 | 0.998 |
| Prompts per day | | | | | | | | |
| 16 | 158 | 0.661 | 0.680 | 0.214 | 0.520 | 0.876 | 0.052 | 0.998 |
| 6 | 156 | 0.628 | 0.637 | 0.197 | 0.498 | 0.791 | 0.017 | 0.968 |
| 5 | 156 | 0.660 | 0.646 | 0.180 | 0.510 | 0.795 | 0.111 | 0.969 |
| 4 | 156 | 0.613 | 0.631 | 0.206 | 0.494 | 0.786 | 0.028 | 0.975 |
| 3 | 154 | 0.704 | 0.665 | 0.192 | 0.567 | 0.809 | 0.000 | 0.967 |
| Feature selection | | | | | | | | |
| All Features | 390 | 0.660 | 0.656 | 0.197 | 0.518 | 0.811 | 0.000 | 0.998 |
| Selected Features | 390 | 0.657 | 0.648 | 0.200 | 0.508 | 0.808 | 0.000 | 0.994 |
| Proportion of own data in training set | | | | | | | | |
| None | 242 | 0.655 | 0.644 | 0.199 | 0.512 | 0.786 | 0.017 | 0.994 |
| 10% | 198 | 0.607 | 0.627 | 0.207 | 0.510 | 0.765 | 0.000 | 0.994 |
| 20% | 170 | 0.673 | 0.674 | 0.181 | 0.532 | 0.819 | 0.258 | 0.998 |
| 30% | 170 | 0.694 | 0.670 | 0.201 | 0.513 | 0.852 | 0.206 | 0.997 |

**Table 6. Linear mixed models of ROC-AUC (lapses).**

| Predictor | Estimate | Standard error | 95% confidence interval | t-statistic | p-value | Sig. |
|---|---|---|---|---|---|---|
| Intercept | 0.677 | 0.039 | 0.601, 0.754 | 17.373 | < 0.001 | *** |
| Prompts per day: 6 | −0.042 | 0.011 | −0.064, −0.020 | −3.749 | < 0.001 | *** |
| Prompts per day: 5 | −0.034 | 0.011 | −0.056, −0.012 | −2.986 | 0.003 | ** |
| Prompts per day: 4 | −0.048 | 0.011 | −0.070, −0.026 | −4.288 | < 0.001 | *** |
| Prompts per day: 3 | −0.014 | 0.011 | −0.036, 0.008 | −1.212 | 0.226 | |
| Feature selection: Selected features | −0.009 | 0.007 | −0.023, 0.005 | −1.215 | 0.224 | |
| Share of own data: 10% | −0.016 | 0.010 | −0.036, 0.003 | −1.615 | 0.106 | |
| Share of own data: 20% | 0.004 | 0.011 | −0.017, 0.025 | 0.384 | 0.701 | |
| Share of own data: 30% | 0.001 | 0.011 | −0.020, 0.021 | 0.057 | 0.954 | |

Reference levels: Prompts per day = 16; Feature selection = All features; Share of own data = None.

Significance levels: *** p < 0.001, ** p < 0.01, * p < 0.05.

**Table 7. Summary of F1-score by algorithm configurations (cravings).**

| | Number of algorithms | Median | Mean | Standard deviation | 25th percentile | 75th percentile | Minimum | Maximum |
|---|---|---|---|---|---|---|---|---|
| Cravings overall | | | | | | | | |
| | 1,424 | 0.400 | 0.410 | 0.328 | 0.048 | 0.649 | 0.000 | 0.997 |
| Prompts per day | | | | | | | | |
| 16 | 292 | 0.470 | 0.465 | 0.332 | 0.141 | 0.745 | 0.000 | 0.997 |
| 6 | 280 | 0.447 | 0.453 | 0.315 | 0.211 | 0.658 | 0.000 | 0.992 |
| 5 | 274 | 0.368 | 0.393 | 0.323 | 0.000 | 0.624 | 0.000 | 0.990 |
| 4 | 290 | 0.361 | 0.379 | 0.329 | 0.000 | 0.647 | 0.000 | 0.987 |
| 3 | 288 | 0.333 | 0.358 | 0.331 | 0.000 | 0.600 | 0.000 | 0.984 |
| Feature selection | | | | | | | | |
| All Features | 712 | 0.410 | 0.415 | 0.322 | 0.117 | 0.646 | 0.000 | 0.997 |
| Selected Features | 712 | 0.400 | 0.404 | 0.334 | 0.000 | 0.650 | 0.000 | 0.997 |
| Proportion of own data in training set | | | | | | | | |
| None | 360 | 0.424 | 0.422 | 0.309 | 0.143 | 0.649 | 0.000 | 0.997 |
| 10% | 356 | 0.419 | 0.412 | 0.315 | 0.123 | 0.633 | 0.000 | 0.997 |
| 20% | 354 | 0.333 | 0.388 | 0.341 | 0.000 | 0.640 | 0.000 | 0.996 |
| 30% | 354 | 0.400 | 0.415 | 0.348 | 0.000 | 0.703 | 0.000 | 0.996 |

**Table 8. Zero-inflated beta-regression model (with random intercepts) of F1-Score (cravings).**

| Predictor | Estimate (logit scale) | Standard error | 95% confidence interval (logit) | z-statistic | p-value | Sig. |
|---|---|---|---|---|---|---|
| Intercept | 0.278 | 0.230 | −0.173, 0.728 | 1.208 | 0.227 | |
| Prompts per day: 6 | −0.142 | 0.056 | −0.253, −0.032 | −2.523 | 0.012 | * |
| Prompts per day: 5 | −0.398 | 0.059 | −0.514, −0.282 | −6.746 | < 0.001 | *** |
| Prompts per day: 4 | −0.346 | 0.059 | −0.461, −0.231 | −5.882 | < 0.001 | *** |
| Prompts per day: 3 | −0.381 | 0.060 | −0.499, −0.262 | −6.304 | < 0.001 | *** |
| Feature selection: Selected features | 0.021 | 0.037 | −0.053, 0.094 | 0.555 | 0.579 | |
| Share of own data: 10% | −0.057 | 0.053 | −0.160, 0.046 | −1.085 | 0.278 | |
| Share of own data: 20% | −0.076 | 0.054 | −0.181, 0.029 | −1.423 | 0.155 | |
| Share of own data: 30% | 0.077 | 0.053 | −0.027, 0.181 | 1.452 | 0.146 | |

Reference levels: Prompts per day = 16; Feature selection = All features; Share of own data = None.

Note: Estimate coefficients are on the logit scale (conditional component only).

This is a zero-inflated beta model with separate processes for zeros and values between 0 and 1.

For interpretable effect sizes (actual differences in predicted values), see pairwise comparisons on the response scale (0–1) in S8 Appendix.

Significance levels: *** p < 0.001, ** p < 0.01, * p < 0.05.

0.500–0.695), and 0.544 (mean 0.541; IQR 0.421–0.676) at 6, 5, 4, and 3 prompts/day, respectively (Table 9). Linear mixed models showed significantly lower ROC-AUCs at all reduced prompt frequencies (all p < .001) (Table 10).

### Research question 2: Feature selection

**Lapses.** Feature selection had minimal impact on lapse prediction. Median F1-scores were similar for models using all predictors (median 0.435; mean 0.404; IQR 0.188–0.620) and selected predictors (median 0.441; mean 0.413; IQR 0.168–0.632) (Table 3). Median ROC-AUCs were likewise similar (0.660 vs 0.657; Table 5). A zero-inflated beta

**Table 9. Summary of ROC-AUC by algorithm configurations (cravings).**

| | N | Median | Mean | Standard deviation | 25th percentile | 75th percentile | Minimum | Maximum |
|---|---|---|---|---|---|---|---|---|
| Cravings overall | | | | | | | | |
| | 1,424 | 0.628 | 0.611 | 0.174 | 0.510 | 0.729 | 0.000 | 1.000 |
| Prompts per day | | | | | | | | |
| 16 | 292 | 0.700 | 0.681 | 0.165 | 0.582 | 0.811 | 0.122 | 0.986 |
| 6 | 280 | 0.652 | 0.641 | 0.148 | 0.575 | 0.736 | 0.000 | 0.983 |
| 5 | 274 | 0.606 | 0.598 | 0.151 | 0.517 | 0.696 | 0.068 | 0.971 |
| 4 | 290 | 0.622 | 0.593 | 0.164 | 0.500 | 0.695 | 0.025 | 1.000 |
| 3 | 288 | 0.544 | 0.541 | 0.201 | 0.421 | 0.676 | 0.000 | 1.000 |
| Feature selection | | | | | | | | |
| All Features | 712 | 0.632 | 0.624 | 0.170 | 0.516 | 0.744 | 0.032 | 1.000 |
| Selected Features | 712 | 0.622 | 0.598 | 0.177 | 0.502 | 0.717 | 0.000 | 0.986 |
| Proportion of own data in training set | | | | | | | | |
| None | 360 | 0.650 | 0.630 | 0.160 | 0.544 | 0.734 | 0.025 | 0.983 |
| 10% | 356 | 0.619 | 0.607 | 0.178 | 0.504 | 0.722 | 0.000 | 1.000 |
| 20% | 354 | 0.632 | 0.606 | 0.177 | 0.503 | 0.728 | 0.032 | 1.000 |
| 30% | 354 | 0.614 | 0.600 | 0.178 | 0.493 | 0.730 | 0.000 | 1.000 |

**Table 10. Linear mixed model of ROC-AUC (cravings).**

| Predictor | Estimate | Standard error | 95% confidence interval | t-statistic | p-value | Sig. |
|---|---|---|---|---|---|---|
| Intercept | 0.714 | 0.018 | 0.678, 0.749 | 39.470 | < 0.001 | *** |
| Prompts per day: 6 | −0.046 | 0.012 | −0.070, −0.023 | −3.812 | < 0.001 | *** |
| Prompts per day: 5 | −0.085 | 0.012 | −0.110, −0.061 | −6.964 | < 0.001 | *** |
| Prompts per day: 4 | −0.088 | 0.012 | −0.111, −0.064 | −7.260 | < 0.001 | *** |
| Prompts per day: 3 | −0.139 | 0.012 | −0.163, −0.115 | −11.489 | < 0.001 | *** |
| Feature selection: Selected features | −0.026 | 0.008 | −0.041, −0.011 | −3.403 | < 0.001 | *** |
| Share of own data: 10% | −0.023 | 0.011 | −0.045, −0.002 | −2.156 | 0.031 | * |
| Share of own data: 20% | −0.023 | 0.011 | −0.044, −0.002 | −2.105 | 0.035 | * |
| Share of own data: 30% | −0.030 | 0.011 | −0.051, −0.008 | −2.726 | 0.006 | ** |

Reference levels: Prompts per day = 16; Feature selection = All features; Share of own data = None.

Significance levels: *** p < 0.001, ** p < 0.01, * p < 0.05p.

model indicated a statistically significant but small improvement for F1-score with feature selection (p = .038), whereas no significant effect was observed for ROC-AUC (p = .224) (Tables 4 and 6).

**Cravings.** For cravings, feature selection was associated with slightly worse performance. Median F1-score decreased from 0.410 (mean 0.415; IQR 0.117–0.646) with all predictors to 0.400 (mean 0.404; IQR 0.000–0.650) with selected predictors, and median ROC-AUC decreased from 0.632 (mean 0.624; IQR 0.516–0.744) to 0.622 (mean 0.598; IQR 0.502–0.717) (Table 7 and Table 9). Formal modelling confirmed statistically significant decrements for ROC-AUC (p < .001), but not F1-score (p = 0.579) (Table 8 and Table 10).

## Research question 3: Outcomes

Outcome differences were more apparent for ROC-AUC than F1-score. Median F1-scores (Table 3 and Table 7) were similar across outcomes (lapses: median 0.436, mean 0.409, IQR 0.180–0.625; cravings: median 0.400, mean 0.410, IQR 0.048–0.649), and the cross-outcome zero-inflated beta model showed no significant difference (p = .061) (Table 11).

**Table 11. Pairwise comparison of F1-Score by outcome; based on zero-inflated beta-regression.**

| Contrast | Estimate | Standard error | z-ratio | p-value |
|---|---|---|---|---|
| Lapses – Cravings | 0.023 | 0.012 | 1.875 | 0.061 |

Model: Zero-inflated beta regression. Estimates shown are for the conditional component only (values between 0 and 1), excluding the zero-inflation process.

Note: Estimates are on the response scale (0–1), showing differences in predicted means.

Positive estimates indicate higher values for the first condition in each contrast; negative estimates indicate higher values for the second condition.

P-values adjusted using Tukey method.

Across all specifications, lapse prediction achieved higher median ROC-AUC (0.659; mean 0.652; IQR 0.514–0.809; Table 5) than craving prediction (0.628; mean 0.611; IQR 0.510–0.729; Table 9). In the cross-outcome linear mixed model (Table 12), this difference was statistically significant (p < .001).

## Research question 4: Proportion of own data

**Lapses.** Median F1-score was lowest when no participant-specific data were included (median 0.286; mean 0.319; IQR 0.000–0.571), increased at 10% (median 0.400; mean 0.384; IQR 0.155–0.607), and was highest at 20−30% (medians 0.500–0.542; means 0.484–0.490; IQRs 0.308–0.667) (Table 3). Zero-inflated beta-regressions indicated that the scores where significantly higher at 20% and 30% of own data included in the training algorithm, relative to none (p = 0.002 and p < .001, respectively) (Table 4).

Median ROC-AUC increased descriptively with more participant-specific data (none: median 0.655; mean 0.644; IQR 0.512–0.786; 30%: median 0.694; mean 0.670; IQR 0.513–0.852), but neither linear mixed models nor adjusted pairwise comparisons indicated statistically significant differences, with all p-values ≥ 0.106 (Table 6; S9 Appendix).

**Cravings.** For cravings, median F1-score was similar for none and 10% participant-specific data (medians 0.424–0.419; means 0.422–0.412; IQRs 0.123–0.649), declined at 20% (median 0.333; mean 0.388; IQR 0.000–0.640), and partially recovered at 30% (median 0.400; mean 0.415; IQR 0.000–0.703) (Table 7). Zero-inflated beta-regressions indicated that none of these differences were statistically significant (all p-values ≥ 0.146) (Table 8).

Median ROC-AUC was highest when no participant-specific data were used (median 0.650; mean 0.630; IQR 0.544–0.734) and lower when any participant-specific data were included (Table 9). Linear mixed models indicated significant decrements at 10% (p = .031), 20% (p = .035), and 30% (p = .006) (Table 10).

## Supplementary metrics

Across specifications, sensitivity (S6 Appendix and S10 Appendix) and specificity (S7 Appendix and S11 Appendix) suggested a consistent trade-off across configurations

**Table 12. Pairwise comparison of ROC-AUC by outcome; based on linear mixed model.**

| Contrast | Estimate | Standard error | t-ratio | p-value |
|---|---|---|---|---|
| Lapses – Cravings | 0.036 | 0.008 | 4.743 | < 0.001 |

Model: Linear mixed model.

Note: Estimates are on the response scale (0–1), showing differences in predicted means.

Positive estimates indicate higher values for the first condition in each contrast; negative estimates indicate higher values for the second condition.

P-values adjusted using Tukey method.

Sensitivity was typically moderate (often ~0.45–0.55), whereas specificity was higher (~0.75–0.85) but more variable. For lapses, lower prompt frequencies were associated with significantly higher sensitivity (all p < .001) alongside substantially lower specificity (again, all p < .001). For cravings, sensitivity differences across prompt frequencies were smaller and in the opposite direction (all significant, at p = .032 to p < .001), while specificity was significantly reduced at 3 and 4 prompts per day, compared to 16 (p = 0.002 and p < .001, respectively).

Together, these supplementary metrics indicate that some configurations maintained or increased sensitivity at the expense of specificity, without necessarily improving overall discrimination.

## Discussion

### Summary of main findings

This study adds to the growing literature on predicting smoking behaviour from EMA data using machine learning algorithms [34,55,66,67]. It does so by systematically examining the effects of prompt frequency, feature selection, prediction target (lapse versus craving), and use of participant-specific data in training on the performance of RF machine learning algorithms to underpin a future smoking cessation JITAI.

Across a large set of algorithm specifications, predictive performance varied substantially between individuals and was, on average, modest when evaluated against commonly used minimum performance thresholds. Median F1-scores for both lapse and craving prediction were often below 0.5, indicating limited balance between precision and recall, while ROC-AUC values were generally above chance but rarely reached levels typically considered strong discrimination. This divergence highlights that many configurations could distinguish higher-risk from lower-risk moments to some extent, yet, at the threshold of 0.5 at least, still struggled to reliably identify true positive events without generating substantial false positives or false negatives. Importantly, these patterns were not driven by a small subset of poorly performing models but reflected broad variability across participants and specifications.

Several consistent trends emerged. For lapse prediction, lower EMA prompt frequencies were associated with higher F1-scores, contrary to expectations that denser sampling would improve performance, whereas ROC-AUC showed no corresponding monotonic improvement and, in some cases, favoured higher prompt frequencies. In contrast, craving prediction showed the expected pattern: reducing prompt frequency led to systematic and statistically significant declines in both F1-score and ROC-AUC. Feature selection had minimal impact on lapse prediction and was associated with slightly worse performance for cravings, suggesting that reducing predictor burden can be achieved with little loss of accuracy for lapses but not necessarily for craving prediction. Finally, the results indicate that the effect of incorporating participant-specific data depends strongly on both the outcome and the metric used to assess performance. For lapses, personalisation improved F1-score but not ROC-AUC, suggesting gains in precision–recall balance without improved discrimination. For cravings, incorporating participant-specific data was associated with poorer performance across both metrics, indicating limited benefit and even potential harm from warm-start personalisation (hybrid-level algorithms) in this setting. Taken together, these findings suggest that while EMA-based machine learning can capture meaningful risk signals, considerations have to be made when it comes to implementing them in real-world JITAIs.

### Practical implications

Taken together, these findings suggest that it may be *possible*, but not guaranteed, to deploy algorithms predicting high-risk moments within a smoking cessation JITAI without imposing excessive user burden, particularly when high-risk moments are operationalised as lapses rather than cravings. Predictive performance was highly variable across individuals and, on average, modest, indicating that such algorithms should be viewed as providing probabilistic risk signals rather than reliable event-level predictions. The comparatively greater difficulty of predicting cravings may be due to the algorithms attempting to predict high cravings *at* the next EMA and lapses *by* the next EMA. As a fluctuating measure, the former may simply be inherently more difficult to predict, especially with longer and more variable gaps between EMAs.

Longer and more variable gaps between EMAs may therefore disproportionately reduce predictive accuracy for cravings compared with lapses.

Although performance generally declined as EMA prompt frequency was reduced, this decline was modest for lapse prediction and depended on the metric used. In particular, lower prompt frequencies were associated with higher F1-scores for lapses, while ROC-AUC showed no corresponding monotonic improvement and, in some cases, favoured higher sampling densities. This divergence highlights that conclusions about "acceptable" performance may depend on which aspect of performance is prioritised. In real-world implementation, it may therefore be necessary to move beyond a fixed classification threshold (e.g., 0.5) and instead select thresholds that explicitly trade off sensitivity and specificity according to intervention goals. This could mean prioritising sensitivity to minimise missed high-risk moments, even at the cost of more false positives. Such threshold tuning could materially change how these algorithms perform in practice relative to the results reported here.

Consistent with the nature of RF models, which intrinsically prioritise the most relevant features, explicit feature selection had little impact on lapse prediction and was associated with small performance decrements for craving prediction. Although we cannot conclude a true absence of an effect, the point estimate of the difference was small. While this does not imply that feature reduction is universally harmless, the very limited reduction in performance across metrics and the overall similarity in retained predictors across algorithm specifications suggests that a relatively small set of psychological and contextual variables captures much of the predictive signal in these data. Given the substantial benefit of reducing survey burden, using a more parsimonious model with fewer psychological and contextual tailoring variables appears to be a justifiable and practical strategy. Nonetheless, concerns about generalisability remain, discussed further below. Previous research indicates that users of a smoking cessation JITAI may not be willing to respond to more than four to five surveys with around 10 items per day [16,19,21,27–30]. Therefore, using one of the lapse risk algorithms using three to six prompts/day and feature selection may enable researchers to develop a smoking cessation JITAI that has the potential to be both acceptable and effective.

Importantly, despite substantial differences in how well the algorithms performed for different individuals, algorithms trained with participant-specific data did not consistently outperform models trained without any of the test participants own data, particularly when looking at ROC-AUC. This has favourable implications in terms of feasibility as it suggests that individualised algorithm training is not necessary to maintain acceptable discriminative ability. However, the benefit of using participant-specific data when evaluating performance using the F1-score suggests that, when it comes to reliably identifying true positive events without generating substantial false positives or false negatives, there may still be a benefit for using hybrid-level algorithms (warm-start personalisation). Additionally, there may still be limitations in terms of generalisability (discussed further below), as the algorithms are trained on individuals who have at least one instance of lapse or high craving and may therefore work less well on those who have none or very few of these events. With these caveats in mind, a group-level lapse-risk algorithm using five prompts per day and feature selection may nevertheless be the best way going forward for a future JITAI based on this research.

Finally, the substantial between-individual variability in performance has important implications for implementation. For some individuals, algorithm performance was consistently poor, suggesting that fully automated decision-making may be inappropriate for all users. One pragmatic approach that may be to combine machine learning outputs with human-designed decision rules or clinician-informed heuristics, such that algorithm-generated risk estimates inform intervention delivery as one of multiple algorithms used by a JITAI. Such hybrid approaches may improve robustness, transparency, and acceptability while mitigating the risks associated with relying on imperfect predictions in high-stakes behavioural contexts.

## Links to previous research

The predictors most consistently selected through RFE in this study, namely cravings, self-efficacy, momentary motivation, cigarette availability, and affective states, are largely consistent with factors identified as important psychological and contextual

variables in prior research identifying predictors of smoking lapses [30,34,67–72]. Location, current activity, and social context were not selected for any feature-reduced algorithms in this study. This aligns with mixed evidence regarding the predictive utility of these contextual features; previous research has found variables relating to these concepts to be key predictors in some models, but not others [34,55,67,68,70,71]. A potential reason why these variables were not selected for any of the feature-reduced algorithms in this study may be how they were operationalised (the variables in this study were fairly fine-grained) and the fact that feature selection was done at the group-level (these variables may still be highly relevant for some individuals). It is also important to note that this study only uses data from the first 10 days of participants' quit attempts. In this period, they may have engaged in non-typical or unsustainably constrained ways to support their quit attempt and study participation, limiting the predictive value of these variables. This interpretation is supported by the relatively low frequency of several physical and social contexts in the data, including "Restaurant/bar," "With friend," and "With relative" (see S12 Appendix). Overall, this suggests that while there is some consistency in the key psychological and contextual predictors of lapse risk, the relevance of certain factors, such as location and social setting, may be more variable, potentially depending on the modelling approach, operationalisations, the dataset, and/or differences between individuals or populations.

The finding that hybrid-level algorithms did not consistently outperform group-level algorithms, and were indeed outperformed by them on some metrics, contrasts with results from previous analyses of machine learning algorithms to predict smoking lapses [67], including the previous study using the same dataset [34]. A likely explanation for this discrepancy is the difference in how the sub-sample of participants' own data that was included in the training set for the hybrid-level algorithms was selected. In the previous studies, hybrid-level algorithms were trained with data that included a randomly selected 20% or 40% of each participant's data. In the present study, to simulate how a hybrid-level algorithm would operate if implemented as part of a real-world smoking cessation intervention, we used the first 10%, 20%, or 30% of each participant's data chronologically. This difference in data selection may have affected the representativeness and diversity of the training data, particularly if, as research using time-varying effect models suggest [68,69,73–78], the predictors of smoking lapses and cravings early on a quit attempt are different than the predictors at later points.

## Strengths, limitations, and directions for future research

This study has several notable strengths. It is, to the authors' knowledge, the first study to assess how EMA sampling frequency, predictor count, and training data composition affect algorithm performance for predicting smoking lapses and cravings. These factors are critical considerations when developing a JITAI, and by systematically examining them, this study can provide a reference for making informed decisions about the necessary trade-offs. Additionally, the use of high-frequency EMA data over a critical 10-day period provides detailed insight into momentary risk factors. Furthermore, the testing of a large number of algorithm configurations and cross-participant validation allows for a nuanced assessment of generalisability and individual-level variability in performance. Finally, the preregistered protocol and open sharing of data and code promote transparency and reproducibility.

There are also several limitations and ways for future research to extend and build on the findings from this study. First, as a secondary analysis of previously collected data, this study used a randomly selected subset of EMAs to explore the impact of reducing sampling frequency. Future studies could assess the ecological validity of the results by experimentally varying prompt frequencies to explore how participants respond to less intensive monitoring regimes and assess the real-world impact of different prompt frequencies on algorithm performance. Although visual inspection of the distributions and time series of all time-varying predictors across participants (see S12 Appendix) did not suggest that reactivity or response blunting was a major concern in this study, future research that experimentally varies prompt frequency would also be well placed to directly test whether and how reactivity influences both response variability and algorithm performance [79]. Second, our analyses focused on short-term prediction, using data from the immediately preceding EMA. Future work could explore models that account for longer-term behavioural patterns or temporal dynamics across the quit attempt to explore whether that would improve predictive performance. Third, the dataset covers only the first 10 days of participants'

smoking cessation attempts. While this period is critical due to the high risk of lapse and relapse, predictors of lapses may change over time [68,69,73–78]. Future research could extend the time period under investigation.

Although algorithm performance varied substantially across individuals, investigating why some participants' smoking behaviour was harder to predict than others' would have required a larger, more diverse sample and is beyond the scope of this study. Future research could examine potential differential performance and algorithmic bias across demographic characteristic to identify person-level characteristics that influence algorithm performance to inform the development of more personalised or stratified approaches. Additionally, as the sample in this study was self-selected, geographically localised, and skewed toward participants willing to engage in intensive EMA protocols, future studies should assess algorithm performance in more diverse and representative populations. This is particularly important given evidence that smoking patterns and lapse predictors may vary by socioeconomic status [80,81], and given that occupation type consistently emerged as a key predictor in this study (see S2 Appendix). While the feature selection procedure identified a relatively stable set of predictors across models, previous work has shown that feature importance can vary substantially across contexts [30,54]. Further work is needed to evaluate the generalisability of these predictors and to better understand their interpretability, including the direction and strength of their association with outcomes. Importantly, relying on a smaller set of predictors may also constrain the range of intervention targets available to a JITAI at any given moment, particularly when some predictors are static or non-modifiable, such as baseline smoking and demographic characteristics. This could limit the diversity of content in the intervention and potentially affect user engagement over time if similar intervention components are repeatedly selected and users get bored of them [7,53,82,83]. Moreover, using fewer predictors may potentially mean omitting personally salient risk factors which may disproportionately affect algorithm performance for some individuals. While the results of this study provide evidence that acceptable predictive performance may be achievable with fewer features under certain conditions, it is important that feature selection decisions in the context of JITAIs consider not only predictive accuracy and participant burden, but also the downstream implications for and consequences of intervention design, including content diversity, sustained user engagement, and effectiveness.

Finally, the algorithms in this study employed a standard decision threshold of 0.5 to classify predicted probabilities when distinguishing between lapses/non-lapses and high cravings/no or low cravings. While intuitively sensible to get accurate predictions and appropriate for exploratory purposes, future work focused on clinical or real-world implementation may benefit from adjusting this threshold to achieve higher F1-scores or optimise either sensitivity or specificity.

## Conclusion

This study examined how EMA prompt frequency, predictor count, outcome definition, and training data composition affect the performance of machine learning algorithms for predicting high-risk moments during smoking cessation. Across specifications, performance was highly variable between individuals and, on average, modest, with ROC-AUC values generally above chance but F1-scores frequently below commonly used thresholds at a fixed decision cut-off. Lapse prediction was more robust than craving prediction, reduced prompt frequency had modest and metric-dependent effects, explicit feature selection had little impact on lapse prediction, and incorporating participant-specific data did not consistently improve performance. Overall, these findings suggest that EMA-based machine learning can capture meaningful risk signals but should be applied cautiously in JITAIs, with careful consideration of outcome choice, performance metric, and decision thresholds. A parsimonious, group-level lapse-risk model using a moderate number of daily prompts may offer a pragmatic starting point, while substantial individual-level variability highlights the potential value of flexible thresholding and hybrid algorithm-rule-based decision strategies in real-world implementation.

## Supporting information

**S1 Appendix. Surveys.**
(DOCX)

**S2 Appendix. List of predictors for each model.**
(DOCX)

**S3 Appendix. Description of observationsfor different algorithm configurations.**
(DOCX)

**S4 Appendix. F1-Score performance summaries by predictor specification and participant.**
(DOCX)

**S5 Appendix. ROC-AUC performance summaries by predictor specification and participant.**
(DOCX)

**S6 Appendix. Sensitivity performance summaries by predictor specification and participant.**
(DOCX)

**S7 Appendix. Specificity performance summaries by predictor specification and participant.**
(DOCX)

**S8 Appendix. F1-Score models and marginal effects.**
(DOCX)

**S9 Appendix. ROC-AUC models and marginal effects.**
(DOCX)

**S10 Appendix. Sensitivity models and marginal effects.**
(DOCX)

**S11 Appendix. Specificity models and marginal effects.**
(DOCX)

**S12 Appendix. Time-varying variables distributions in full dataset.**
(DOCX)

## Acknowledgments

Thank you to all participants for their invaluable contributions to this research.

## Author contributions

**Conceptualization:** Corinna Leppin, Jamie Brown, Claire Garnett, Olga Perski.

**Data curation:** Corinna Leppin, David Simons, Olga Perski.

**Formal analysis:** Corinna Leppin.

**Funding acquisition:** Jamie Brown.

**Investigation:** Corinna Leppin, Dimitra Kale, Tosan Okpako, Olga Perski.

**Methodology:** Corinna Leppin, Olga Perski.

**Project administration:** Dimitra Kale, Olga Perski.

**Supervision:** Jamie Brown, Claire Garnett, Olga Perski.

**Validation:** Corinna Leppin, Olga Perski.

**Visualization:** Corinna Leppin, David Simons.

**Writing – original draft:** Corinna Leppin.

**Writing – review & editing:** Corinna Leppin, Jamie Brown, Claire Garnett, Dimitra Kale, Tosan Okpako, David Simons, Olga Perski.

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
