## [Decision Letter · Decision Letter 0]

4 Nov 2025

PONE-D-25-43992Optimising supervised machine learning algorithms predicting cigarette cravings and lapses for a smoking cessation just-in-time adaptive intervention (JITAI)PLOS ONE

Dear Dr. Leppin,

Thank you for submitting your manuscript to PLOS ONE. After careful consideration, we feel that it has merit but does not fully meet PLOS ONE’s publication criteria as it currently stands. Therefore, we invite you to submit a revised version of the manuscript that addresses the points raised during the review process.

**Please see below comments from myself and two expert reviewers regarding suggestions for you to consider if you choose to resubmit.**

We look forward to receiving your revised manuscript.

Kind regards,

Jesse T. Kaye, PhD

Academic Editor

PLOS ONE

**Journal Requirements:**

1. When submitting your revision, we need you to address these additional requirements. Please ensure that your manuscript meets PLOS ONE's style requirements, including those for file naming. The PLOS ONE style templates can be found at https://journals.plos.org/plosone/s/file?id=wjVg/PLOSOne_formatting_sample_main_body.pdf and https://journals.plos.org/plosone/s/file?id=ba62/PLOSOne_formatting_sample_title_authors_affiliations.pdf 2. Please note that PLOS One has specific guidelines on code sharing for submissions in which author-generated code underpins the findings in the manuscript. In these cases, we expect all author-generated code to be made available without restrictions upon publication of the work. Please review our guidelines at https://journals.plos.org/plosone/s/materials-and-software-sharing#loc-sharing-code and ensure that your code is shared in a way that follows best practice and facilitates reproducibility and reuse. 3. We note that the grant information you provided in the ‘Funding Information’ and ‘Financial Disclosure’ sections do not match.  When you resubmit, please ensure that you provide the correct grant numbers for the awards you received for your study in the ‘Funding Information’ section. 4. Thank you for stating in your Funding Statement: This analysis is part of a project funded by Cancer Research UK (PRCRPG-Nov21\100002; https://www.cancerresearchuk.org/funding-for-researchers/our-funding-schemes/prevention-and-population-research-programme-awards). CG is supported by a National Institute for Health and Care Research (NIHR) Advanced Fellowship (#302923; https://www.nihr.ac.uk/career-development/research-career-funding-programmes). OP is supported by a Marie Skłodowska-Curie Postdoctoral Fellowship from the European Union (Grant Agreement number: 101065293; https://marie-sklodowska-curie-actions.ec.europa.eu/actions/postdoctoral-fellowships). Views and opinions expressed are however those of the author(s) only and do not necessarily reflect those of the European Union. The European Union cannot be held responsible for them. The funders had no role in study design, data collection and analysis, decision to publish, or preparation of the manuscript.  Please provide an amended statement that declares *all* the funding or sources of support (whether external or internal to your organization) received during this study, as detailed online in our guide for authors at http://journals.plos.org/plosone/s/submit-now. Please also include the statement “There was no additional external funding received for this study.” in your updated Funding Statement. Please include your amended Funding Statement within your cover letter. We will change the online submission form on your behalf. 5. Thank you for stating the following in the Competing Interests section: I have read the journal's policy and the authors of this manuscript have the following competing interests: OP and JB act as unpaid scientific advisors to the Smoke Free app. CG, DK, CL, TO, and DS have no competing interests to declare.  We note that one or more of the authors are employed by a commercial company.  a. Please provide an amended Funding Statement declaring this commercial affiliation, as well as a statement regarding the Role of Funders in your study. If the funding organization did not play a role in the study design, data collection and analysis, decision to publish, or preparation of the manuscript and only provided financial support in the form of authors' salaries and/or research materials, please review your statements relating to the author contributions, and ensure you have specifically and accurately indicated the role(s) that these authors had in your study. You can update author roles in the Author Contributions section of the online submission form. Please also include the following statement within your amended Funding Statement. “The funder provided support in the form of salaries for authors [insert relevant initials], but did not have any additional role in the study design, data collection and analysis, decision to publish, or preparation of the manuscript. The specific roles of these authors are articulated in the ‘author contributions’ section.”If your commercial affiliation did play a role in your study, please state and explain this role within your updated Funding Statement.  b. Please also provide an updated Competing Interests Statement declaring this commercial affiliation along with any other relevant declarations relating to employment, consultancy, patents, products in development, or marketed products, etc.   Within your Competing Interests Statement, please confirm that this commercial affiliation does not alter your adherence to all PLOS ONE policies on sharing data and materials by including the following statement: "This does not alter our adherence to  PLOS ONE policies on sharing data and materials.” (as detailed online in our guide for authors http://journals.plos.org/plosone/s/competing-interests) . If this adherence statement is not accurate and  there are restrictions on sharing of data and/or materials, please state these. Please note that we cannot proceed with consideration of your article until this information has been declared. Please include both an updated Funding Statement and Competing Interests Statement in your cover letter. We will change the online submission form on your behalf. 6. We noted in your submission details that a portion of your manuscript may have been presented or published elsewhere. “A previous study using the same data has been published in PLOS Digital Health (https://doi.org/10.1371/journal.pdig.0000594). The previous study demonstrated that it is possible to accurately predict smoking lapses and cravings using very high-density EMA and sensor data. In contrast, this manuscript addresses how such models might be adapted for real-world implementation by exploring trade-offs between data collection burden and prediction performance.”Please clarify whether this [conference proceeding or publication] was peer-reviewed and formally published. If this work was previously peer-reviewed and published, in the cover letter please provide the reason that this work does not constitute dual publication and should be included in the current manuscript. 7. We note that this data set consists of interview transcripts. Can you please confirm that all participants gave consent for interview transcript to be published? If they DID provide consent for these transcripts to be published, please also confirm that the transcripts do not contain any potentially identifying information (or let us know if the participants consented to having their personal details published and made publicly available). We consider the following details to be identifying information:- Names, nicknames, and initials- Age more specific than round numbers- GPS coordinates, physical addresses, IP addresses, email addresses- Information in small sample sizes (e.g. 40 students from X class in X year at X university)- Specific dates (e.g. visit dates, interview dates)- ID numbers Or, if the participants DID NOT provide consent for these transcripts to be published:- Provide a de-identified version of the data or excerpts of interview responses- Provide information regarding how these transcripts can be accessed by researchers who meet the criteria for access to confidential data, including:a) the grounds for restrictionb) the name of the ethics committee, Institutional Review Board, or third-party organization that is imposing sharing restrictions on the datac) a non-author, institutional point of contact that is able to field data access queries, in the interest of maintaining long-term data accessibility.d) Any relevant data set names, URLs, DOIs, etc. that an independent researcher would need in order to request your minimal data set. For further information on sharing data that contains sensitive participant information, please see: https://journals.plos.org/plosone/s/data-availability#loc-human-research-participant-data-and-other-sensitive-data If there are ethical, legal, or third-party restrictions upon your dataset, you must provide all of the following details (https://journals.plos.org/plosone/s/data-availability#loc-acceptable-data-access-restrictions):a) A complete description of the datasetb) The nature of the restrictions upon the data (ethical, legal, or owned by a third party) and the reasoning behind themc) The full name of the body imposing the restrictions upon your dataset (ethics committee, institution, data access committee, etc)d) If the data are owned by a third party, confirmation of whether the authors received any special privileges in accessing the data that other researchers would not havee) Direct, non-author contact information (preferably email) for the body imposing the restrictions upon the data, to which data access requests can be sent 8. If the reviewer comments include a recommendation to cite specific previously published works, please review and evaluate these publications to determine whether they are relevant and should be cited. There is no requirement to cite these works unless the editor has indicated otherwise.

**Additional Editor Comments:**

This manuscript was reviewed by two experts who agreed that the study addresses important methodological issues in the optimization of ML algorithms for JITAIs. Figuring out evidence-based strategies to strike the right balance between minimizing data collection burden while maintaining model accuracy is an essential step towards development of JITAIs that truly have potential for real-world adoption. The manuscript is well written and thoroughly describes the complex methods and analytic approaches in a straightforward manner that is understandable to tobacco treatment researchers without ML expertise, while including sufficient detail for researchers with ML/JITAI domain expertise. Further, I appreciate the authors' commitment to implementing transparent and open science practices throughout the manuscript. I believe that this manuscript will be a great contribution to the literature, which will be strengthened by addressing the thoughtful and constructive comments provided by the two reviewers. Please pay particular attention to the comments identified as more major (e.g., item variability, moderation by smoking recency) and provide a justification for your rationale if you do not address any minor comments in the revised manuscript, if you choose to resubmit.

Reviewers' comments:

Reviewer's Responses to Questions

**Comments to the Author**

1. Is the manuscript technically sound, and do the data support the conclusions?

Reviewer #1: Yes

Reviewer #2: Partly

2. Has the statistical analysis been performed appropriately and rigorously? 

Reviewer #1: Yes

Reviewer #2: Yes

3. Have the authors made all data underlying the findings in their manuscript fully available?

Reviewer #1: Yes

Reviewer #2: Yes

4. Is the manuscript presented in an intelligible fashion and written in standard English?

Reviewer #1: Yes

Reviewer #2: Yes

5. Review Comments to the Author

**Reviewer #1:** This manuscript describes an important project that aimed to build a set of risk prediction models for smoking behavior and craving among individuals undergoing a quit attempt. The authors used dense EMA sampling during the first 10 days post-quit to build models that differed systematically on characteristics relevant to participant burden. The manuscript is clearly written with a concise/direct introduction and thorough methods. Throughout, the authors demonstrate thoughtful decision-making regarding eventual clinical use/implementation that guided their choices in model-building. I appreciate the authors' commitment to open science, particularly the nice explanation of changes from preregistration and the availability of analysis scripts, which were helpful for me to review.

Overall, this manuscript seems clear, impactful, and rigorous. I have several suggestions for manuscript improvement, but no concerns about the quality of the science.

1. The information about selection of algorithm(s) appears conflicting across elements of the paper. In the "Selection of algorithm types" section, you state you only tested one algorithm, but the analysis scripts suggest you tested multiple algorithms. You mention that RF also outperformed other algorithms. In the analysis scripts (e.g., in "18_ML*.R"), you also describe selecting using auROC (e.g., "although the SVM had slightly higher AUC, it had unacceptable sensitivity. therefore, the RF was selected"), but my understanding is that F1 was your primary optimizing/performance metric. Could you please clarify this process?

2. It was not initially clear to me (until I got to the Results, in fact) that you were predicting craving and lapses in separate models. This was originally a concern that I removed because I thought you were combining these heterogeneous outcomes into a single outcome of "high risk moments" - I am glad you are not combining them, but some improved clarify on this point earlier in the intro/methods would be helpful.

3. Relatedly, I know it’s currently in the Appendix, but I wonder if the CLMM results/performance that are currently in Tables 4 and 3 should be replaced with tables stratified by outcome. Lapses and cravings feel like qualitatively different outcomes to me, and although you note that most patterns were consistent across outcomes (e.g., line 406), I think displaying the results stratified by outcome feels appropriate.

4. In your inclusion criteria, what defines "regularly smoking"?

5. Was there any confirmation of smoking status at the in-person baseline visit, or only confirmation of quit at follow-up (i.e., using CO)?

6. How many participants were excluded for insufficient EMA adherence (from the full dataset to the preprocessed sub-set that you used)? Relatedly, given that many of your models used fewer than the full 16 prompts, why not include individuals with < 60% EMA adherence?

7. How/why did you select 7 as the cut-off for the dichotomous split of craving data?

8. (From lines 459-461) How would lower specificity help explain why ROC-AUC values would be comparatively higher than F1 scores? Wouldn’t low specificity values pull ROC-AUC (but NOT F1) down, as F1 doesn’t consider specificity? I may just be misunderstanding what you wrote, but this is not clear to me.

9. Given the accurate and impactful conclusion you draw that using one of the lapse risk algorithms using 5 prompts per day and feature selection could be feasible and acceptable (e.g., lines 506-508; lines 516-518), I wonder if it would be worth presenting that algorithm specification’s performance in the Discussion.

10. An important “risk” of using fewer features in a risk prediction algorithm is that you limit the available features from which you can map to JITAI tools/modules. For example, if you only have 8 features, and some are demographic/static (e.g., age, occupation) and thus unable to be modified, you are limited to ~6 target areas in which to intervene. When looking at the features contributing to risk for a single observation (e.g., using something like local Shapley values) to inform the specific intervention recommendation in the moment, you might run into issues with only being able to offer one tool or type of tool (e.g., methods to manage craving), which could get annoying quickly for a patient. Relatedly, given the variability in performance across people, it may be that although the algorithms worked well overall with fewer features, perhaps there are consequences for individual people - if a feature is not important to them, that matters more when there are only 8 total features compared to dozens. This is not to say it’s not important to consider participant burden! But I wonder if it’s worth discussing this trade-off/risk given your focus on implementation.

11. Did you notice any patterns in the performance across participants based on demographic characteristics (e.g., race, gender, income, education, employment)? I know you mentioned employment emerged as a consistently important predictor, and of course this sample size is too small to make between-participant conclusions, but I'm curious whether there were any indications of algorithmic bias. I will completely understand if this is impossible to discuss given the sample size!

12. An alternative explanation for why contextual variables were not selected here is that they were not important in THIS time window (10 days post-quit) and/or that there was not sufficient variability in these variables in this period. Individuals may have exhibited more homogenous behavior immediately post-quit - e.g., being around more people to distract themselves, avoiding triggering people, not going to bars, etc. - that might not be sustainable longer-term.

13. Not required, but I think you can make a stronger case for this project’s findings of group vs. hybrid given your choice of including the first 10-30% of a participant’s own data - you note the previous study selected the data randomly, which means they could be “predicting” someone’s behavior from their own future behavior. In my eyes, your methods on this front are more rigorous than the previous work using these data.

**Reviewer #2:** Thank you for the opportunity to review “Optimising supervised machine learning algorithms predicting cigarette cravings and lapses for a smoking cessation just-in-time adaptive intervention (JITAI).” In an effort to minimize participant burden for a future JITAI, this paper examines the effect of several study design factors on the performance of a random forest algorithm to predict smoking lapse and craving among people who are trying to quit smoking cigarettes. Results from this study found high variability in the performance of the algorithm with better performance for smoking lapse compared to cigarette craving. Reducing the number of EMAs had little effect on performance. Surprisingly, including the participant’s own data was associated with worse performance. This work is important to intervention development, given the increasing interest in using machine learning to tailor treatments. This paper provides much needed data to inform the balance of participant burden with sufficient data to identify lapse risk. Overall, the paper is well-written; specifically, the introduction outlines the relevant research and rationale for the current project very well. However, several major concerns are noted with the methods for this project that warrant further elaboration and discussion before publication.

Major comments

Overall:

• Of primary concern is the use of the participant’s own data in the training set. Participant data could extend out to the third day of the quit attempt. As the authors note, lapses may or may not have occurred within this period (line 325). Key within-person variables in the algorithm including affect, craving, and self-efficacy likely have very different predictive utility, depending on if the person has recently smoked or not. For example, low irritability symptoms when the person has not smoked recently may be an indicator that they are not experiencing severe withdrawal, therefore they are less likely to lapse. However, low irritability when the person has smoked recently would likely be less predictive of lapse. It is unclear how recency of smoking during the first three days was included in the algorithm or in the definition of a lapse. Not including recency of smoking may explain the surprising result that the participant’s own data did not improve performance. Additionally, approximately a third of the sample was not included in the hybrid models. Could this reduction in sample size, confounding individual difference variables (e.g., cigarette dependence), and/or not including the recency of smoking explain the worse algorithm performance among certain participants? If possible, additional analyses may be helpful for identifying if any of these variables predict worse algorithm performance. Additionally, these limitations need greater consideration and expansion within the discussion. There is brief mention of some of these issues (line 553-554), but this should be discussed in more detail.

• 16 daily EMA prompts is high and may lead to reactivity where participants provide little variability in their responses. Was there low variability in any of the variables included in the algorithm? Does this explain the poor performance of the algorithm for some of the participants? Particularly those that showed little change in their responses?

Methods:

• The rationale for dichotomizing craving into a low/high variable is unclear and may have impacted results. The variable from this 11-point scale may be ordinal or continuous, therefore, creating a binary variable could potentially reduce power and possibly worsen algorithm performance. However, dichotomizing this variable may have been necessary to predict discrete high risk craving events. Or it may have been naturally split within the distribution, but this is not clear in the paper. If craving cannot be treated as continuous, would it be possible to examine low, moderate, and high craving occasions? Some work suggests strong peaks in craving are most predictive of lapse, therefore merging more moderate cravings with high cravings could obfuscate results (line 201-204).

• When selecting the number of EMA prompts what would happen if a participant did not have enough data for one of the levels (e.g., a participant only completing 4 of the 16 prompts on a given day)? Were the participants only represented in lower levels (e.g., 3 or 4 prompts used)? If so, this may lead to unequal ns which could bias results if only highly compliant participants are represented in certain levels. Similarly, all event-based EMAs were included. Does having more/less event-based EMAs affect algorithm performance (lines 228-241)?

Discussion:

• I believe the conclusion that individualized algorithms are not necessary is too definitive. It appears there may be other artifacts within the data that may have worsened performance which should be discussed as another possible explanation (line: 512-518).

Minor comments

Introduction:

• What does past work suggest is the length/number of needed prompts for a warm-start to improve algorithm performance (line 102-105)?

Methods:

• Please define the specific eligibility requirement for “regularly smoking cigarettes” (e.g., minimum cigarettes per day; line 141).

• Please include measures from the baseline survey (line 150).

• The measures section (lines 173-189) is difficult to follow. It was unclear which variables were included as predictors in the algorithm or baseline variables or both.

• Does past hour nicotine use include cigarettes, non-combustible tobacco, or both (line 187)?

• Please clarify what is meant by the threshold value of 0.5 (line 214).

Results:

• Please include how many participants were excluded due to insufficient EMA adherence.

Discussion:

• The discussion would benefit from additional guidance on what point estimates suggest acceptable algorithm performance and how different variables (e.g., number of EMA prompts) fall within the general range for acceptable/unacceptable performance.

6. PLOS authors have the option to publish the peer review history of their article (what does this mean?). If published, this will include your full peer review and any attached files.

Reviewer #1: **Yes:** Gaylen Fronk

Reviewer #2: No

---

## [Author Response · Author response to Decision Letter 1]

1 Mar 2026

Response to Reviewer Comments

We have gone through and checked that our manuscript adheres to the PLOS ONE style and formatting requirements.

We have made all author-generated code and underlying data on Github (https://github.com/OlgaPerski/EMASENS_ML/tree/main) and the Open Science Framework (https://osf.io/tnu72/).

The sections have been updated and now match.

This analysis is part of a project funded by Cancer Research UK (PRCRPG-Nov21\100002; https://www.cancerresearchuk.org/funding-for-researchers/our-funding-schemes/prevention-and-population-research-programme-awards). CG is supported by a National Institute for Health and Care Research (NIHR) Advanced Fellowship (#302923; https://www.nihr.ac.uk/career-development/research-career-funding-programmes). OP is supported by a Marie Skłodowska-Curie Postdoctoral Fellowship from the European Union (Grant Agreement number: 101065293; https://marie-sklodowska-curie-actions.ec.europa.eu/actions/postdoctoral-fellowships). Views and opinions expressed are however those of the author(s) only and do not necessarily reflect those of the European Union. The European Union cannot be held responsible for them. The funders had no role in study design, data collection and analysis, decision to publish, or preparation of the manuscript.

The statement has been amended.

I have read the journal's policy and the authors of this manuscript have the following competing interests: OP and JB act as unpaid scientific advisors to the Smoke Free app. CG, DK, CL, TO, and DS have no competing interests to declare.

We note that one or more of the authors are employed by a commercial company. [CL1.1]

Two of the authors act as unpaid scientific advisors for a commercial company, but none of the authors are employed by or receive salaries or any other remuneration from any commercial companies. The Competing Interest Statement has been updated to clarify that.

6. We noted in your submission details that a portion of your manuscript may have been presented or published elsewhere. “A previous study using the same data has been published in PLOS Digital Health (https://doi.org/10.1371/journal.pdig.0000594). The previous study demonstrated that it is possible to accurately predict smoking lapses and cravings using very high-density EMA and sensor data. In contrast, this manuscript addresses how such models might be adapted for real-world implementation by exploring trade-offs between data collection burden and prediction performance.”

A prior publication using the same data was peer-reviewed and formally published in PLOS Digital Health. That study established proof of concept by demonstrating that smoking lapses and cravings can be accurately predicted using very high-density EMA and sensor data.

The present manuscript does not constitute dual publication because it addresses a substantively different research question with a distinct and unique contribution. Rather than maximising predictive accuracy using intensive data collection, this work focuses on real-world implementation by systematically examining trade-offs between data collection burden and model performance. Specifically, it evaluates how prediction models can be adapted when data density is reduced to levels that are feasible and sustainable in practical clinical or public health settings. These analyses, results, and implications were not examined in the prior publication and represent a novel and complementary contribution beyond the earlier proof-of-concept findings.[JB2.1]

7. We note that this data set consists of interview transcripts. Can you please confirm that all participants gave consent for interview transcript to be published?

If they DID provide consent for these transcripts to be published, please also confirm that the transcripts do not contain any potentially identifying information (or let us know if the participants consented to having their personal details published and made publicly available). We consider the following details to be identifying information:

- Names, nicknames, and initials

- Age more specific than round numbers

- GPS coordinates, physical addresses, IP addresses, email addresses

- Information in small sample sizes (e.g. 40 students from X class in X year at X university)

- Specific dates (e.g. visit dates, interview dates)

- ID numbers

Or, if the participants DID NOT provide consent for these transcripts to be published:

- Provide a de-identified version of the data or excerpts of interview responses

- Provide information regarding how these transcripts can be accessed by researchers who meet the criteria for access to confidential data, including:

a) the grounds for restriction

b) the name of the ethics committee, Institutional Review Board, or third-party organization that is imposing sharing restrictions on the data

c) a non-author, institutional point of contact that is able to field data access queries, in the interest of maintaining long-term data accessibility.

d) Any relevant data set names, URLs, DOIs, etc. that an independent researcher would need in order to request your minimal data set.

For further information on sharing data that contains sensitive participant information, please see: https://journals.plos.org/plosone/s/data-availability#loc-human-research-participant-data-and-other-sensitive-data

If there are ethical, legal, or third-party restrictions upon your dataset, you must provide all of the following details (https://journals.plos.org/plosone/s/data-availability#loc-acceptable-data-access-restrictions):

a) A complete description of the dataset

b) The nature of the restrictions upon the data (ethical, legal, or owned by a third party) and the reasoning behind them

c) The full name of the body imposing the restrictions upon your dataset (ethics committee, institution, data access committee, etc)

d) If the data are owned by a third party, confirmation of whether the authors received any special privileges in accessing the data that other researchers would not have

e) Direct, non-author contact information (preferably email) for the body imposing the restrictions upon the data, to which data access requests can be sent

We have not submitted any data consisting of interview transcripts or any analysis or results based on interview data. All data in this paper is anonymised EMA data that has been openly shared via GitHub (https://github.com/OlgaPerski/EMASENS_ML/tree/main/data)

Thank you and we have noted this.

Additional Editor Comments:

This manuscript was reviewed by two experts who agreed that the study addresses important methodological issues in the optimization of ML algorithms for JITAIs. Figuring out evidence-based strategies to strike the right balance between minimizing data collection burden while maintaining model accuracy is an essential step towards development of JITAIs that truly have potential for real-world adoption. The manuscript is well written and thoroughly describes the complex methods and analytic approaches in a straightforward manner that is understandable to tobacco treatment researchers without ML expertise, while including sufficient detail for researchers with ML/JITAI domain expertise. Further, I appreciate the authors' commitment to implementing transparent and open science practices throughout the manuscript. I believe that this manuscript will be a great contribution to the literature, which will be strengthened by addressing the thoughtful and constructive comments provided by the two reviewers. Please pay particular attention to the comments identified as more major (e.g., item variability, moderation by smoking recency) and provide a justification for your rationale if you do not address any minor comments in the revised manuscript, if you choose to resubmit.

Thank you for these kind comments. We have updated the manuscript in line with the expert reviews. Furthermore, during revision, we identified an error in the specification of the positive class for threshold-dependent performance metrics (F1-score, sensitivity, specificity), arising from the use of a deprecated option for event-level setting. This resulted in non-lapses/low cravings being treated as the positive class. We corrected the event-level specification, re-computed all performance metrics, and re-ran the analyses of the performance metrics. Following correction, F1-scores, sensitivity, and specificity were lower across algorithm specifications, whereas ROC-AUC values changed comparatively little. All results, tables, and interpretations have been updated accordingly.

Reviewers' comments:

Reviewer's Responses to Questions

Comments to the Author

1. Is the manuscript technically sound, and do the data support the conclusions?

Reviewer #1: Yes

Reviewer #2: Partly

2. Has the statistical analysis been performed appropriately and rigorously?

Reviewer #1: Yes

Reviewer #2: Yes

3. Have the authors made all data underlying the findings in their manuscript fully available?

Reviewer #1: Yes

Reviewer #2: Yes

4. Is the manuscript presented in an intelligible fashion and written in standard English?

Reviewer #1: Yes

Reviewer #2: Yes

5. Review Comments to the Author

Reviewer #1: This manuscript describes an important project that aimed to build a set of risk prediction models for smoking behavior and craving among individuals undergoing a quit attempt. The authors used dense EMA sampling during the first 10 days post-quit to build models that differed systematically on characteristics relevant to participant burden. The manuscript is clearly written with a concise/direct introduction and thorough methods. Throughout, the authors demonstrate thoughtful decision-making regarding eventual clinical use/implementation that guided their choices in model-building. I appreciate the

---

## [Decision Letter · Decision Letter 1]

15 Apr 2026

PONE-D-25-43992R1Optimising supervised machine learning algorithms predicting cigarette cravings and lapses for a smoking cessation just-in-time adaptive intervention (JITAI)PLOS One

Dear Dr. Leppin,

Thank you for submitting your manuscript to PLOS ONE. After careful consideration, we feel that it has merit but does not fully meet PLOS ONE’s publication criteria as it currently stands. Therefore, we invite you to submit a revised version of the manuscript that addresses the points raised during the review process. We invite you to submit a revised version of this manuscript with very minor revisions - to addresses the issue of clarifying your sample size with greater transparency as recommended by reviewer 1. I leave it to your discretion how to incorporate this clarification in the manuscript (e.g., tables, manuscript body). I expect that you will be able to address this issue with relative ease and I anticipate being able to accept this manuscript for publication in PLOS ONE without further need to send out again to reviewers. Please ensure that you cite your prior PLOS Digital Health manuscript with 1-2 sentences describing the distinct aims. I want to echo the reviewers sentiments that this is a very rigorous and well executed study. I applaud your transparency in noting the coding error that you detected during the revision process and thoroughly addressing in this revision. The manuscript addresses very complex analytic questions with rigor and humility, not overstating inferences or conclusions throughout. I look forward to following your continued work in this area and future contributions to the literature.

We look forward to receiving your revised manuscript.

Kind regards,

Jesse T. Kaye, PhD

Academic Editor

PLOS One

Journal Requirements:

Reviewers' comments:

Reviewer's Responses to Questions

**Comments to the Author**

1. If the authors have adequately addressed your comments raised in a previous round of review and you feel that this manuscript is now acceptable for publication, you may indicate that here to bypass the “Comments to the Author” section, enter your conflict of interest statement in the “Confidential to Editor” section, and submit your "Accept" recommendation.

Reviewer #1: (No Response)

Reviewer #2: All comments have been addressed

2. Is the manuscript technically sound, and do the data support the conclusions?

Reviewer #1: Yes

Reviewer #2: Yes

3. Has the statistical analysis been performed appropriately and rigorously? 

Reviewer #1: Yes

Reviewer #2: Yes

4. Have the authors made all data underlying the findings in their manuscript fully available?

Reviewer #1: Yes

Reviewer #2: Yes

5. Is the manuscript presented in an intelligible fashion and written in standard English?

Reviewer #1: Yes

Reviewer #2: Yes

6. Review Comments to the Author

Reviewer #1: I am very satisfied with how my comments were addressed and believe the manuscript is greatly improved. Thank you for your thorough and thoughtful responses. I have two remaining comments:

1. After reading the revisions to the manuscript, I am left a bit confused about sample size. I have two points of confusion:

a. First, in the "Design" and "Sample Description" sections, you note that N = 37. In contrast, "Data pre-processing" notes that you excluded 8 people for low EMA adherence from an original N of 46, which would mean 38 remaining participants. Please help explain this discrepancy.

b. Second, and perhaps partially primed by the insightful note from Reviewer 2 about the possible role of limited sample size/less data in the hybrid models, I feel the manuscript would benefit from more clarification about the differing sample sizes across model contexts. Given the requirements for inclusion that differ across contexts (e.g., at least one observation of each class for group models, at least one observation of each class in test in hybrid models [meaning - I think - that they needed an observation in each class after the first 1-3 days], different positive classes across craving and lapse outcomes), I think sample size should be more concretely defined. One possible solution is to add a column to the results tables that lists how many participants were included in a given model context (and/or the number of observations [EMAs] included), though I will defer to the authors' preference as to how to include this information most clearly. Regardless, knowing the actual sample sizes rather than only the starting N would improve my ability as a reader to understand model performance & implications.

2. In lines 282 - 284 (clean copy), it currently reads as if event-based EMA prompts were only included for the model specifications that used reduced numbers of prompts. My understanding is that this is not true, so please clarify these lines slightly. Alternatively, if this is true, please justify why event-based EMA prompts were not included in the 16-EMA models.

Additionally, some typographical errors remain throughout and will be important to address during copy-editing.

Reviewer #2: Thank you for the opportunity to review the revised manuscript “Optimising supervised machine learning algorithms predicting cigarette cravings and lapses for a smoking cessation just-in-time adaptive intervention (JITAI).” Overall, the authors have addressed my comments from the prior submission. I also commend the authors for noting the error in their modeling, correcting it, and bringing it to our attention (that’s good science!). Findings now suggest these algorithms only modestly detect high-risk lapse and craving moments, with substantial individual variability. Typically, metrics fell below the standard for acceptable performance. I can imagine this was a difficult issue to grapple with at this stage in the project, but the authors exemplify responsible research with their approach. I appreciate the straightforward reporting of these findings with the discussion focused on the difficulties in this work that may have impacted model performance and areas for improvement in future work. I believe this work continues to provide a thoughtful contribution to this literature appropriate for publication in PLoS One.

7. PLOS authors have the option to publish the peer review history of their article (what does this mean?). If published, this will include your full peer review and any attached files.

Reviewer #1: **Yes:** Gaylen Fronk

Reviewer #2: No

---

## [Author Response · Author response to Decision Letter 2]

17 Apr 2026

Response to Reviewers

1. We invite you to submit a revised version of this manuscript with very minor revisions - to addresses the issue of clarifying your sample size with greater transparency as recommended by reviewer 1. I leave it to your discretion how to incorporate this clarification in the manuscript (e.g., tables, manuscript body). I expect that you will be able to address this issue with relative ease and I anticipate being able to accept this manuscript for publication in PLOS ONE without further need to send out again to reviewers.

• We thank the reviewers and the editor for their kind and helpful comments. We have incorporated their recommendations as detailed below.

2. Please ensure that you cite your prior PLOS Digital Health manuscript with 1-2 sentences describing the distinct aims.

• A section describing the distinct aims has now been added: “The current study aimed to train and test a random forest algorithm for a future smoking cessation JITAI. A previously published proof-of-concept study used a series of supervised machine learning algorithms to demonstrate that smoking lapses and cravings can be accurately predicted using high-density EMA and sensor data (Perski et al., 2024). This study extends that work using the same data, focusing on real-world implementation by systematically examining trade-offs between data collection burden and model performance. It explores how reducing the sampling frequency and the number of variables and varying the proportion of the test participants’ own data in the training set affects the algorithm’s ability to predict high-risk moments, operationalised either as lapses or as instances of high cravings and predicted in separate algorithms, during a smoking cessation attempt. The aim is to evaluate how prediction models can be adapted when data density is reduced to levels that are feasible and sustainable in practical clinical or public health settings.” (lines 123-135)

3. After reading the revisions to the manuscript, I am left a bit confused about sample size. I have two points of confusion:

• First, in the "Design" and "Sample Description" sections, you note that N = 37. In contrast, "Data pre-processing" notes that you excluded 8 people for low EMA adherence from an original N of 46, which would mean 38 remaining participants. Please help explain this discrepancy.

• Thank you for spotting this omission! This difference is now explained: “The analytic sample comprises n = 37 adult people who smoke from in and around London during the first 10 days of their quit attempt. Initial pre-processing had left a sample of n=38 participants, but one further participant was excluded from the analytic sample after because they had no instances of either high cravings or lapses and therefore did not meet the requirement of having at least one observation of each outcome class in each outcome set.” (lines 407-410)

• Second, and perhaps partially primed by the insightful note from Reviewer 2 about the possible role of limited sample size/less data in the hybrid models, I feel the manuscript would benefit from more clarification about the differing sample sizes across model contexts. Given the requirements for inclusion that differ across contexts (e.g., at least one observation of each class for group models, at least one observation of each class in test in hybrid models [meaning - I think - that they needed an observation in each class after the first 1-3 days], different positive classes across craving and lapse outcomes), I think sample size should be more concretely defined. One possible solution is to add a column to the results tables that lists how many participants were included in a given model context (and/or the number of observations [EMAs] included), though I will defer to the authors' preference as to how to include this information most clearly. Regardless, knowing the actual sample sizes rather than only the starting N would improve my ability as a reader to understand model performance & implications.

• Thank you for pointing this out – this information was previously only buried in the supplemental files. We have now added this information in the sample description: “Algorithms using a proportion of the participants’ own data in the training set had lower sample sizes as only participants with at least one observation in each outcome class in the testing set were included. For algorithms predicting lapses, algorithms using 10% of participants’ own data used data from 20 participants, while algorithms using 20% and 30% of participants’ own data used data from 17 participants. For algorithms predicting cravings, algorithms using 10% of participants’ own data used data from all 37 participants, while algorithms using 20% and 30% of participants’ own data used data from 36 participants.” (lines 425-431)

• Details on the number of observations for the different models are now also included in S3 appendix: “Versions of this table restricted to participants with at least one lapse and to participants included in the different algorithm configurations using a subset of the test participant’s own data in the training set are available in S3 Appendix.” (lines 436-438)

4. In lines 282 - 284 (clean copy), it currently reads as if event-based EMA prompts were only included for the model specifications that used reduced numbers of prompts. My understanding is that this is not true, so please clarify these lines slightly. Alternatively, if this is true, please justify why event-based EMA prompts were not included in the 16-EMA models.

• Thank you for pointing this out. We have now modified the section to reflect that event-based EMAs were also included in the 16-prompt algorithms: “All event-based EMA reports (i.e., when a participant reported smoking outside of a prompt) were included in all algorithms, regardless of the number of prompts per day tested.” (lines 280-282)

---

## [Decision Letter · Decision Letter 2]

24 Apr 2026

Optimising supervised machine learning algorithms predicting cigarette cravings and lapses for a smoking cessation just-in-time adaptive intervention (JITAI)

PONE-D-25-43992R2

Dear Corinna Leppin,

We’re pleased to inform you that your manuscript has been judged scientifically suitable for publication and will be formally accepted for publication once it meets all outstanding technical requirements.

Kind regards,

Dr Anna Tovmasyan

Academic Editor

PLOS One

Additional Editor Comments:

Only a few minor issues left that could be addressed at proofs stage. P. 21, line 254 - please insert the closing bracket. P. 42, line 614 - please format "Practical implications" as a subheading. P. 44, line 661 - please remove duplicated commas. P. 46 - the section is titled "Strengths, limitations, and directions for future research", but the section does not actually name any limitations. Please re-word the text in this section to clearly identify which limitations the study has before recommending further research.

Reviewers' comments:

Reviewer's Responses to Questions

**Comments to the Author**

1. If the authors have adequately addressed your comments raised in a previous round of review and you feel that this manuscript is now acceptable for publication, you may indicate that here to bypass the “Comments to the Author” section, enter your conflict of interest statement in the “Confidential to Editor” section, and submit your "Accept" recommendation.

Reviewer #1: All comments have been addressed

Reviewer #2: All comments have been addressed

2. Is the manuscript technically sound, and do the data support the conclusions?

Reviewer #1: Yes

Reviewer #2: Yes

3. Has the statistical analysis been performed appropriately and rigorously? 

Reviewer #1: Yes

Reviewer #2: Yes

4. Have the authors made all data underlying the findings in their manuscript fully available?

Reviewer #1: Yes

Reviewer #2: (No Response)

5. Is the manuscript presented in an intelligible fashion and written in standard English?

Reviewer #1: Yes

Reviewer #2: Yes

6. Review Comments to the Author

Reviewer #1: (No Response)

Reviewer #2: (No Response)

7. PLOS authors have the option to publish the peer review history of their article (what does this mean?). If published, this will include your full peer review and any attached files.

Reviewer #1: **Yes:** Gaylen Fronk

Reviewer #2: No

---

## [Editor Report · Acceptance letter]

PONE-D-25-43992R2

PLOS One

Dear Dr. Leppin,

I'm pleased to inform you that your manuscript has been deemed suitable for publication in PLOS One. Congratulations! Your manuscript is now being handed over to our production team.

Kind regards,

on behalf of

Dr. Anna Tovmasyan

Academic Editor

PLOS One